# Transient Receptor Potential Channel Expression Signatures in Tumor-Derived Endothelial Cells: Functional Roles in Prostate Cancer Angiogenesis

**DOI:** 10.3390/cancers11070956

**Published:** 2019-07-08

**Authors:** Michela Bernardini, Alessia Brossa, Giorgia Chinigò, Guillaume P. Grolez, Giulia Trimaglio, Laurent Allart, Audrey Hulot, Guillemette Marot, Tullio Genova, Aditi Joshi, Virginie Mattot, Gaelle Fromont, Luca Munaron, Benedetta Bussolati, Natalia Prevarskaya, Alessandra Fiorio Pla, Dimitra Gkika

**Affiliations:** 1Univ. Lille, Inserm, U1003-PHYCEL-Physiologie Cellulaire, F-59000 Lille, France; 2Laboratory of Excellence, Ion Channels Science and Therapeutics, Université de Lille, F-59655 Villeneuve d’Ascq, France; 3Department of Life Sciences and Systems Biology, University of Torino, 10123 Turin, Italy; 4Department of Molecular Biotechnology and Health Sciences, Molecular Biotechnology Centre, University of Torino, 10126 Turin, Italy; 5Univ. Lille, Institut Français de Bioinformatique, bilille, F-59000 Lille, France; 6Univ. Lille, Inria, CHU Lille, EA 2694-MODAL-Models for Data Analysis and Learning, F-59000 Lille, France; 7Univ. Lille, CNRS, Institut Pasteur de Lille, UMR 8161, F-59000 Lille, France; 8Inserm UMR 1069, Université de Tours, 37000 Tours, France

**Keywords:** tumor angiogenesis, calcium channel, migration, TRP, prostate cancer

## Abstract

*Background*: Transient receptor potential (TRP) channels control multiple processes involved in cancer progression by modulating cell proliferation, survival, invasion and intravasation, as well as, endothelial cell (EC) biology and tumor angiogenesis. Nonetheless, a complete TRP expression signature in tumor vessels, including in prostate cancer (PCa), is still lacking. *Methods:* In the present study, we profiled by qPCR the expression of all TRP channels in human prostate tumor-derived ECs (TECs) in comparison with TECs from breast and renal tumors. We further functionally characterized the role of the ‘*prostate-associated*’ channels in proliferation, sprout formation and elongation, directed motility guiding, as well as in vitro and in vivo morphogenesis and angiogenesis. *Results*: We identified three ‘*prostate-associated*’ genes whose expression is upregulated in prostate TECs: TRPV2 as a positive modulator of TEC proliferation, TRPC3 as an endothelial PCa cell attraction factor and TRPA1 as a critical TEC angiogenic factor in vitro and in vivo. *Conclusions*: We provide here the full TRP signature of PCa vascularization among which three play a profound effect on EC biology. These results contribute to explain the aggressive phenotype previously observed in PTEC and provide new putative therapeutic targets.

## 1. Introduction

Transient receptor potential (TRP) channels form a 28-member superfamily of nonselective channels mostly permeable to both monovalent and divalent cations, which is subgrouped into six main families, namely, TRPC (canonical), TRPV (vanilloid), TRPM (melastatin), TRPP (polycystin), TRPML (mucolipin), and TRPA (ankyrin). Despite their structural similarities, TRP channels are polymodal molecular sensors and are activated by a range of external stimuli (including temperature, light, sound, chemicals, and touch) and changes in local microenvironment [1]. These observations suggest that the physiologically relevant stimulus for any given TRP is governed by the specific cellular context, which dramatically changes during carcinogenesis [2].

Indeed, TRP-activated signaling pathways have profound effects on a variety of physiological and pathological processes, including cancer [3,4]. Accumulating evidence demonstrates that the development of some cancers involves ion channels, thereby classifying these diseases as a special type of ‘*channelopathy*’, called ‘*oncochannelopathy*’ [2,4,5]. Several studies have demonstrated that TRP channels control multiple aspects of cancer progression at all stages and modulate Ca^2+^-regulated pathways such as cell proliferation and survival, migration and angiogenesis [2,6]. It is therefore not surprising that the expression of some TRP channels is altered during tumor growth and metastasis, and consequently a TRP expression signature could characterize the phenotype of a tumor, including prostate cancer (PCa) [7]. Several studies have identified and proposed TRP channels as new prognostic and therapeutic targets for PCa [8,9,10,11,12]. In particular, we proposed that at least five TRP members (namely, TRPM8, TRPV6, TRPC6, TRPV2 and TRPA1) could be of particular interest in PCa therapy since their expression and function define and modulate specific stages of PCa progression (for a review see [5]).

Nevertheless, little is known about the role of TRP channels in PCa angiogenesis, even though the crucial role of vascularization in tumor growth, invasion into surrounding tissues and metastasis is well studied. Indeed, as for other cancers, tumor angiogenesis is a well-recognized hallmark of aggressive PCa progression [13]. Immunohistochemical staining on normal and tumor tissues from the prostate showed that the endothelial cell (EC) population is enriched in PCa tissues compared to that in normal tissues, whereas the EC population decreases upon castration and gradually recovers with time [14]. A growing number of studies implicate TRP channels in EC biology, as well as in tumor angiogenesis. Several members of the TRP superfamily (i.e., TRPC1, TRPC4, TRPC6, TRPM7, TRPV1, TRPV4 and TRPM8) appear to be heterogeneously expressed in different EC types and have a key role during multiple functions in ECs, including cell proliferation, survival, motility and tubulogenesis in vitro [15,16,17]. However, until now, most available data focus on the function of a single channel rather a global view of the regulation of all TRP channels in PCa development and angiogenesis [16].

The definition of a specific expression pattern is of particular interest, as it is now evident that specific stages of cancer progression, such as angiogenesis, are not caused by mutations in the *trp* genes but rather by altered expression of the wild-type protein [4]. Indeed, the aim of this study is to profile the expression of TRP channels during PCa angiogenesis and to identify the specific molecular modulators of this process, thereby identifying novel therapeutic targets. We have therefore analyzed the complete expression profile of all *trp* channels in prostate tumor-derived ECs (TEC). To verify the PCa specificity of the molecular candidates identified, we also profiled *trp* expression signatures in ECs derived from breast and renal tumors. We identified four ‘*prostate-associated*’ genes whose expression is significantly and specifically deregulated in prostate TECs relative to normal ECs. Most importantly, we functionally characterized the role of the ‘*prostate-associated*’ *trp* channels (i.e., *trpa1*, *trpv2*, and *trpc3*) upregulated in TECs in the modulation of key biological processes occurring during angiogenesis, such as EC proliferation, motility, regulation of tubulogenesis in vitro and angiogenic sprouting in vivo.

## 2. Results

### 2.1. Identification of the trp Expression Signature Associated with PTEC

To define a complete ‘*TRP signature*’ for the PCa endothelium, we performed gene expression profiling of TRP channels by means of real-time qPCR in TECs from the prostate and compared the profiles to those of breast and renal tumors. For this purpose, we designed primers targeting the functional region for all members of the TRP channel superfamily (*ankyrin 1*, *trpa1*; *canonical 1-7*, *trpc1-7*; *melastatin 1-8*, *trpm1-8*; *vanilloid 1-6*, *trpv1-6*; *polycystin 2-3*, *trpp2-3*; and *mucolipin 1-3*, *trpm1-3*).

We screened three primary human TEC primary cell lines derived from the prostate (PTEC1-3, corresponding to three different patients), which were previously characterized in our laboratory [18]. We also screened TECs from breast and renal tumors (designated BTEC and RTEC, respectively) [19,20]. Expression profiles of TECs were compared to those of several normal human ECs, namely, human umbilical vein ECs (HUVECs), human microvascular ECs from derma (HMECs), glomerular ECs from the kidney (GECs), and the commercially available human prostatic microvascular ECs (HPrMEC, ScienCell Research Laboratories, Carlsbad, CA, USA).

Differential analysis by the ΔΔ*C_T_* method and fold change analysis of the expression values resulting from the qPCR screening were used to select ‘*prostate-associated*’ genes that are deregulated during PCa angiogenesis (Figure 1c).

The ΔΔ*C_T_* method was applied to calculate the relative *trp* expression level in both normal ECs and TECs. ΔΔ*C_T_* values in TECs were normalized to different normal ECs: PTEC1-3, BTEC and RTEC expression profiles were compared to those of HPrMEC, HMEC and GEC respectively (Figure 1a). As shown in the heatmap (Figure 1a), the complete profiling of *trp* channel expression revealed several genes that are deregulated between normal ECs and TECs. Notably, compared to those in HPrMEC six genes are significantly deregulated in PTEC1-3 (Figure 1b), and among these, *trpa1* is the only upregulated gene (Figure 1d, Appendix A). To identify a specific profile for PCa vascularization, we compared the expression profiles of TECs from PCa to those of TECs from breast and renal cancers. Four mRNA (i.e., *trpc1*, *trpc4*, *trpm1*, and *trpml1*) are downregulated in all TECs compared to that in normal ECs; moreover, *trpm7* is downregulated in PTEC and RTEC but upregulated in BTEC (Figure 1a, Appendix A). Based on differential statistical analysis, *trpa1* is the only ‘*prostate-associated*’ gene (Figure 1c). In fact, in RTEC, *trpa1* expression is barely detectable, and *trpa1* is downregulated relative to that in renal GECs; in BTEC, *trpa1* is expressed at low levels, being downregulated relative to that in HPrMEC and upregulated relative to that in other normal ECs (HUVEC, HMEC and GEC) (Figure 1d). We further analyzed the expression profiles in PTEC, in comparison with HPrMEC (Figure 1e). Notably, twelve genes display at least 2-fold change in their expression relative to those in HPrMEC in at least two patient-derived PTEC (Figure 1e, blue and red dots, Figure 1c). Among these genes, *trpa1*, *trpv2*, and *trpc3* are specifically upregulated in PTEC but not in the other TECs (Figure 1d).

In particular, *trpa1*, already identified by differential analysis (Figure 1b,c), is strongly upregulated in PTEC1-3 relative to that in HPrMEC (51.72-fold increase) (Figure 1d, Appendix A), as well as relative to that in HUVEC and HMEC (where channel expression is barely detectable) and GEC. *trpv2* is upregulated by greater than 2-fold (2.47-fold increase) in PTEC1 and PTEC3 (PTEC1,3) relative to that in HPrMEC (Figure 1d, Appendix A) and is overall markedly upregulated compared to that in other normal ECs while; however, in other TECs (RTEC and BTEC) and in normal HMEC, *trpv2* expression is barely detectable (Figure 1d). *trpc3* is strongly upregulated in PTEC1,3 (39.38-fold increase) relative to that in normal HPrMEC (Figure 1d, Appendix A), and this up-regulation is also consistent relative to that in HUVEC and HMEC (Figure 1d). *trpc3* is downregulated in RTEC relative that in the normal GEC, while in BTEC *trpc3* is downregulated relative to that in HMEC but upregulated relative to that in HUVEC and HPrMEC (Figure 1d). The only *trp* gene showing a specific downregulation in PTEC is *trpc6*, being 34-fold downregulated compared to that in HPrMEC (Figure 1d, Appendix A). No significant change in *trpc6* expression is observed between RTEC and GECs, while BTEC display upregulation of *trpc6* relative to that in all normal ECs (HUVEC, HMEC, andGEC) but not to HPrMEC (Figure 1d).

These data therefore highlight four channels, namely, *trpa1*, *trpv2*, *trpc3* and *trpc6,* as ‘*prostate-associated*’ (Figure 1c), displaying at least 2-fold change in expression compared to that in normal ECs (Figure 1e, red dots); additionally, these genes show differential regulation in PTEC compared with those in other TECs (Figure 1d).

### 2.2. TRPA1, TRPV2, TRPC3 and TRPC6 Expression in PCa Patients

Given that we identified *trpa1*, *trpv2*, *trpc3* and *trpc6* as ‘*prostate-associated*’ genes that are deregulated in PTEC compared to that in normal HPrMEC, we further studied their expression at the protein level in vivo by immunohistochemistry on PCa tissues (Figure 2a,b) from ten patients (see materials and methods section) and in vitro by western blotting on proteins extracted from PTEC1-3 (Appendix A). Immunohistochemical analysis highlighted TRPA1 expression in human PCa tissues, showing positive TRPA1 staining in intratumoral ECs (Figure 2a_aii_). In particular, endothelial expression of TRPA1 was observed in tumor areas in all patient tissues (*n* = 10/10), with negative staining on cancer cells in 8 out of 10 patient samples (Figure 2a_aii_). In the normal areas of the tissues, we observed focal positive staining in epithelial cells with diffuse staining in ECs in all patients (Figure 2a_ai_). We quantified TRPA1 expression by western blotting in PTEC and compared it to the expression in HPrMEC confirming TRPA1 upregulation in all PTEC with a mean increase by 1.78-fold (Appendix A).

Immunohistochemical analysis in PCa patient samples confirmed TRPV2 overexpression in intratumoral ECs, showing positive expression in seven of 10 patient samples (Figure 2a_bii_), with variable expression in tumor cells (Figure 2a_bii_). However, TRPV2 expression was not detected in ECs in histologically normal areas (*n* = 0/10) (Figure 2a_bi_). The expression pattern at the protein level was confirmed by western blotting, showing TRPV2 overexpression in PTECs compared to that in HPrMEC (Appendix A), in accordance with the mRNA levels in the different PTEC (Appendix A).

Immunohistochemistry did not confirm TRPC3 expression patterns in PTEC mRNA profiling, revealing negative TRPC3 staining in ECs in all normal areas (*n* = 10/10 patients) (Figure 2a_ci_), but only one tumor area (*n* = 1/10) showed faint positive TRPC3 staining in ECs (Figure 2a_cii_). Nevertheless, *trpc3* already shows a certain degree of variability since it is overexpressed in two PTEC (PTEC1,3) out of three. The overexpression pattern detected by mRNA profiling was confirmed by western blotting, showing an upregulation in all PTECs (Appendix A).

As previously stated, we also identified *trpc6* as a ‘*prostate-associated*’ downregulated gene. Immunohistochemistry for TRPC6 confirmed the qPCR data, as expression of the channel was observed in ECs in normal areas (*n* = 6/10) (Figure 2a_di_), while a negative staining was obtained in ECs in all tumor areas (*n* = 10/10) (Figure 2a_dii_). TRPC6 expression was absent in cancer cells in 7/10 patient samples (Figure 2a_dii_). However, PTEC showed variable expression of TRPC6 at the protein level, as western blotting analysis showed no change in TRPC6 expression at the protein level in PTEC relative to that in HPrMEC (Appendix A).

Overall, except for TRPC3, the results obtained are consistent at the protein level and the mRNA level determined by qPCR screening, thereby establishing ‘*prostate-associated*’ TRP channels as valid candidates to study at the functional level by defining their role in TEC physiology. We have further studied functional expression of TRPA1, TRPV2 and TRPC3 since up-regulation in mRNA and protein levels do not necessarily correspond to increased channel activity. Ca^2+^ imaging experiments on the three primary cells PTEC 1, 2 and 3 using the agonists of TRPA1 (Figure 2c), TRPV2 (Figure 2d) and TRPC3 (Figure 2e) showed a significant increased Ca^2+^ influx as compared to HPrMEC in accordance with the over-expression of these channels in PTECs.

### 2.3. ‘Prostate-Associated’ TRPV2 Enhances EC Viability

We functionally characterized the role of the three *“prostate-associated”* overexpressed TRP channels (namely, *trpa1*, *trpv2*, and *trpc3*) in TECs. We first assessed the effect of the overexpression of the selected TRP channels in cell viability, a key process in stalk cells during angiogenic sprouting [6,7]. For this purpose, we used HMECs as a cellular model to study the function and role of the selected ‘*prostate-associated*’ channels (*trpa1*, *trpv2*, and *trpc3*) since they exhibit a low or barely detectable expression as shown by qPCR (Figure 1d). For all experiments, we verified the overexpression and activity of the channels by western blotting and Ca^2+^ imaging experiments, respectively, using the following channel activators: AITC for TRPA1; LPC for TRPV2; and OAG for TRPC3 (Appendix A). Of the three TRP channels analyzed, only TRPV2 significantly increased basal HMEC viabiity by approximately 55% relative to control cells, as evaluated by MTS assays (Figure 3a).

To study the functional role of the channels in PTEC, we immortalized PTEC (hTERT PTEC) by retroviral transfection with telomerase reverse transcriptase (hTERT). To validate the new cell model, we performed flow cytometric analysis to confirm endothelial markers expression (Appendix A). Similarly, to primary PTEC, hTERT PTEC expressed the following endothelial markers: Endoglin, Type I membrane glycoprotein (CD105), Platelet EC adhesion molecule (CD31, PECAM-1), and Vascular Endothelial Growth Factor Receptor1 (VEGFR1, Appendix A). CD105 and CD31 expression was also confirmed at the mRNA level by qPCR (Appendix A). Phenotype maintenance was confirmed by flow cytometry analysis of endothelial markers expression at passage 7 (Appendix A). In addition, androgen receptor (AR) expression was detected in hTERT PTEC (Appendix A), as previously reported for primary PTEC [18]. In line with this last study, hTERT PTEC cell lines were resistant to sorafenib (Appendix A), an anti-angiogenic drug currently used in the treatment of several cancers.

We further validated *trp* channel expression and function in hTERT PTEC as compared with PTEC3 (from which the cells were immortalized), HPrMEC and HMEC. Immortalized hTERT PTEC showed equal expression of *trpa1*, *trpv2*, *trpc3* and *trpc6* as compared with PTEC3 cells (Figure 4a). hTERT PTEC protein expression was validated by immunoblot and compared with HPrMEC (Figure 4b). Induction of the endogenous TRPV2, TRPA1 and TRPC3/TRPC6 channels activity in PTEC was functionally validated by Ca^2+^ imaging experiments. Stimulation of hTERT PTEC cells with the agonists of TRPA1, TRPV2 and TRPC3 (20 µM AITC, 10 µM LPC and 50 µM OAG respectively) induced sustained Ca^2+^ influx and increased significantly the maximum amplitude of response in comparison with that in HPrMEC, which barely express the channel proteins (Figure 4c).

Similarly HMEC stimulation with AITC and LPC show barely no Ca^2+^ responses; however OAG promoted a significant increase in [Ca^2+^]_i_ probably due to the non specific activity of OAG and possible activation of TRPC6 (Figure 4c, Appendix A). Altogether, these data show that hTERT PTEC are suitable as a PTEC model for the study of the selected TRP candidates in cell processes important for EC physiology, such as cell viability, migration, chemotaxis and tubulogenesis. We next assessed whether the activation of the endogenous *trpa1*, *trpv2*, and *trpc3* channels affected the viability of hTERT PTEC. In line with the overexpression results in HMEC (Figure 3a), only treatment with 10 µM LPC (a TRPV2 agonist) induced an increase of approximately 45.5% in hTERT PTEC viability compared to that under control conditions (Figure 3b), confirming the role of endogenous TRPV2 as a positive modulator of EC proliferation. Silencing of TRPV2 expression significantly reduced availability in hTERT PTEC, under both basal conditions and LPC stimulation (Figure 3c). The role of TRPV2 in cell proliferation was confirmed by means of BrDU assay (Appendix A). Downregulation of TRPV2 expression was verified by Ca^2+^ imaging experiments 48 hours after siRNA transfection after stimulating control and TRPV2-silenced (siTRPV2) hTERT PTEC with 10 µM LPC (Figure 3e,f). Mean TRPV2 downregulation of 44.3% was verified 48 hours after siRNA transfection by means of western blotting (Figure 3d).

### 2.4. ‘Prostate-Associated’ TRPA1 Expression Promotes EC Migration

Cell migration is one of the key steps involved in sprouting angiogenesis that enables endothelial tip cells to function as motile guidance structures that dynamically extend filopodia to explore signals in the tumor microenvironment for directional vessel growth [21]. We thus investigated the role of the four ‘*prostate-associated*’ channels in EC migration. For this purpose, we overexpressed the channels in HMECs to analyze their effect on EC motility by means of scratch wound healing assays. Among the three candidates, overexpression of TRPA1 significantly increased migration in HMEC compared to that in non-transfected cells (Figure 5a). The increase in migration rate was significant from 2 h onward and peaked after 8 hours in TRPA1-overexpressing HMEC compared to that in control cells.

We further investigated the endogenous role of TRPA1 in hTERT PTEC motility, as this channel is strongly up regulated in PTEC compared to that in normal ECs (Figure 1 and Figure 2). To achieve this goal, we tested the effect of TRPA1 silencing in hTERT PTEC by wound healing experiments (Figure 5b). Compared to siCNTRL transfection, down regulation of TRPA1 expression in PTEC by means of siRNA transfection induced a significant decrease of 23% in cell migration (siTRPA1) (Figure 5b). Moreover, TRPA1 inhibition by treatment with the channel inhibitor (20 µM HC030031) induced a 41% decrease in migration in control hTERT PTEC (siCNTRL) but not in TRPA1-silenced hTERT PTEC (siTRPA1) (Figure 5b). Next, endogenous functional expression of TRPA1 in hTERT PTEC and HMEC was tested by Ca^2+^ imaging. [Ca^2+^]_i_ significantly increased in hTERT PTEC but not in HMEC following stimulation with the TRPA1 channel agonist AITC (20 µM) (Figure 5d). On the other hand, silencing of TRPA1 expression in hTERT PTEC significantly reduced AITC-mediated [Ca^2+^]_i_ by 3.26-fold (Figure 5d). Similarly, treatment with two different concentrations (10 µM and 20 µM) of the TRPA1 inhibitor HC030031 dramatically decreased AITC-induced channel activity in hTERT PTEC (Figure 5e). Mean TRPA1 downregulation of 36.8% was verified via western blotting in PTEC 48 h after siRNA transfection (Figure 5c).

### 2.5. ‘Prostate-Associated’ TRPC3 Promotes PCa Cell Attraction

TECs have been shown to exert a chemoattractive effect on cancer cell, thus guiding cancer metastasis [22]. We thus considered a possible role of the three *‘prostate-associated’* channels that are upregulated in PTEC (*trpa1*, *trpv2* and *trpc3*) in the cross-talk between TECs and PCa cells. In this regard, we showed that hTERT PTEC exert a significantly stronger attractive effect on PCa cells (PC3) in co-culture experiments than do normal ECs (HMEC) (Figure 6a). Moreover, we overexpressed TRPA1, TRPV2 and TRPC3 channels in HMEC and investigated the effect on PC3 cell attraction. TRPC3 overexpression in HMEC induced a significant increase of 219% in PC3 cell attraction compared to that in control cells, mimicking the effect of PTEC that endogenously overexpress the channel (Figure 6a). However, TRPA1 and TRPV2 overexpression in HMECs did not alter PC3 cell attraction compared to that in control HMEC in co-culture experiments (Figure 6a central panel), supporting a specific role for TRPC3 in PCa cell attraction.

We further explored the role of TRPC3 in PCa cell attraction by chemoattraction assays in the presence of hTERT PTEC, TRPC3-overexpressing HMEC, or control HMEC conditioned media (Figure 6b). We observed that compared to HMEC conditioned medium, hTERT PTEC conditioned medium (24-h conditioning) increased PC3 cell migration speed (Figure 6b,c, left panel). Moreover, compared to control HMEC conditioned media, the conditioned medium from TRPC3-overexpressing HMEC significantly promoted PC3 migration via enhancing the directional persistence index but not the speed (Figure 6b,c, central and right panels). These data, together with the results from the transwell migration experiments, confirm that TRPC3 expression is correlated with PCa cell attraction by TECs.

### 2.6. TRPA1 Promotes Vascular Network Formation and Angiogenic Sprouting both In Vitro and In Vivo

The results described above indicate that TRPA1 is a modulator of PTEC migration, raising the question of whether this channel also affects tubulogenesis or in vivo angiogenesis. To investigate this point, we tested TRPA1 and other TRP candidates for their ability to organize into capillary-like structures in vitro. Interestingly, among all the *‘prostate-associated’* channels, a significant increase in the capillary-like structure formation was observed in TRPA1-overexpressing HMEC relative to that in control cells (Figure 7a, Appendix A). In particular, for TRPA1 overexpression, we observed a strong and significant increase in the number of *master segments* and a concomitantly significant decrease of number of *isolated segments*, two important parameters of EC organization during network formation (Figure 7a, right panel). The role of endogenous TRPA1 was subsequently studied on hTERT PTEC capillary-like structure formations: both TRPA1 inhibitor HC030031 treatment or TRPA1 down regulation strongly and significantly decreases the number of *master segments* and a significantly increase the number of *isolated segments* (Figure 7a, right panels). These results suggest that in addition to playing an important role in cell migration, TRPA1 exerts a marked effect on in vitro EC morphogenesis, which is one of the key features required for new vessel formation (Figure 7a).

To further confirm the role of TRPA1 in vessel morphogenesis, we performed in vivo experiments by using the well-established in vivo angiogenesis model of postnatal mouse retina. Endogenous TRPA1 is diffusely expressed in postnatal mouse retina but clearly co-localizes with CD31 staining, indicating that TRPA1 is expressed in ECs (Figure 7b). Moreover, TRPA1 expression was detected by qPCR in CD31-positive cells isolated from the whole retina, confirming its expression in retinal ECs (Figure 7c). Subretinal injection of AITC or HC030031 did not alter the overall development of the vascular plexus 3 days after treatment but disturbed the vascular front. Indeed, the front of the growing vasculature appeared more uneven after injection of the TRPA1 agonist AITC in the subretinal space. This was illustrated by a significant increase of 26.8% in the length of the sprouts (Figure 7e, upper graph, white bars). In addition, when vascular sprouts were counted and classified in terms of their length (<70 µm, 70–150 µm or >150 µm), the number of longer sprouts (>70 µm) was increased after AITC injection (Figure 7f), confirming the irregular growth front of the retinal vascular plexus when TRPA1 is activated. In contrast, the retina vascular front was straighter when the TRPA1 inhibitor HC030031 was injected in the vitreous, confirmed by a 14.1% reduction in the length of the vascular sprouts (Figure 7e, lower graph, white bars) and by an increase in the number of the shorter sprouts (<70 µm) and a decrease in the number of longer sprouts (Figure 7f, lower panel). These data demonstrate an important role of TRPA1 in vascular remodeling (as demonstrated by capillary-like morphology in Matrigel), as well as in vascular sprouting in vivo in the retina. Therefore, we detected an important role of TRPA1 (Figure 7a) by overexpression or inhibition of its basal activity using the TRPA1 blocker HC030031, (Figure 7d) further supporting the role of TRPA1 in vascular remodeling.

The significant role of TRPA1 in sprouting angiogenesis intrigued us to evaluate a possible activity of this channel in chemoattraction. In order to verify this hypothesis, we performed migration transwell assays stimulating hTERT PTEC cells in the presence of 20 µM AITC in the lower compartment. Figure 8 clearly shows a chemoattractive effect in cell stimulated with TRPA1 agonist as compared to control (Figure 8a). We next evaluated whether this chemoattractive role was due to specific activation of TRPA1 on migrating cells.

For this purpose we performed Ca^2+^ imaging experiments on hTERT PTEC forming capillary-like structures. Interestingly we observed a specific distribution of TRPA1 activity in hTERT PTEC plated on Matrigel/collagen substrate as compared to 2D culture dishes (Figure 8b). 20 µM AITC induced a strong increase in [Ca^2+^]i only in cells engaged in migration (here called “tip-like” cells to recall “tip” cell organization in vessel morphogenesis, Figure 8b,c). These data are in agreement with the role of TRPA1 in vessel morphogenesis and cell migration suggesting an active role for the channel in sprouting angiogenesis and tip cells activation.

## 3. Discussion

The present study provides a full TRP channel expression signature in TECs from PCa by shedding light on their expression profile and the role of the most prominent TRP channels in tumor angiogenesis. We identified four ‘*prostate-associated*’ genes (i.e., *trpa1*, *trpv2*, *trpc3,* and *trpc6*), which were deregulated in previously isolated and characterized PTEC [18] compared to those in ECs derived from the healthy prostate. Among them, three ‘*prostate-associated*’ channels (TRPA1, TRPV2, and TRPC3) were overexpressed during PCa angiogenesis and showed proangiogenic activity by exerting effects on the principal EC properties during angiogenesis, particularly EC proliferation, to support sprout formation, directed motility guiding, sprout elongation and in vitro and in vivo morphogenesis regulation.

The study of global changes in *trp* channel expression patterns during physiopathological transitions and disease progression is important for understanding the development of many pathologies, such as cancer [16]. Moreover, a molecular *trp* signature of a particular cancer could represent a promising prognostic tool for several cancer types. In this regard, several studies have been recently performed, and microarray-assisted expression profiling has identified several ion channel genes (including *trp* channels) to be differentially expressed in tumor tissues relative to those in normal tissues and to contribute to various extents to pathophysiological hallmarks in breast cancer [23], lung adenocarcinoma [24], glioma [25] or pancreatic ductal adenocarcinoma [26].

Here, we show that TRPV2 is a positive modulator of TEC viability, since LPC-mediated activation of endogenous TRPV2 in PTEC resulted in increased viability and proliferation, and downregulation of the channel by siRNA transfection reverted this effect. These data underline a basal role for TRPV2 even in the absence of agonist stimulation. We confirm the different patterns of *trpv2* mRNA expression obtained by qPCR in PCa vessels by immunohistochemistry on tissues obtained from radical prostatectomies in patients (7 cases out of 10). Even though TRPV2 transcript expression has been reported in human pulmonary artery [27] and umbilical vein [28] ECs, its role in these cells is not described. In circulatory organs, TRPV2 is mainly expressed in smooth muscle cells and participates in mechanosensation [28]. Our study showed, for the first time, a role for TRPV2 in tumor angiogenesis by clearly demonstrating the channel as a positive modulator of EC viability and proliferation. TRPV2 has been previously indirectly implicated in angiogenesis since activation of the channel in fibroblasts by its synthetic cannabinoid agonist (O-1821) was correlated with altered angiogenesis in a mouse model of arthritis [29]. The proangiogenic role we described here for TRPV2 adds another aspect to the tumorigenic properties of TRPV2, since this channel has been positively correlated with tumor progression to the castration-resistant phenotype in PCa [11] and with tumor grade and stage in bladder cancer [30]. In both these urogenital cancers, basal and agonist-induced activity of TRPV2 increased the migratory potential of cancer cells. Lysophospholipids and adrenomedullin stimulate insertion of the channel into the plasma membrane, and the subsequent TRPV2-mediated Ca^2+^ influx increases invasiveness of tumor cells via the direct regulation of key proteases such as MMP2, MMP9, and cathepsin B [11,31,32]. However, in this study, we demonstrated that TRPV2 activation can result in different physiological effects depending on the cell type, since activation of endothelial TRPV2 enhanced cell proliferation rather than cell motility. Because agonists of TRPV2 are implicated not only in tumor progression but also in angiogenesis [33,34,35,36], we can speculate that TRPV2 activity could be targeted in both TECs and normal ECs. The use of TRPV2 as an antitumor target is certainly an exciting prospect in drug research but should be considered with caution for cancers other than PCa.

The tumor microenvironment, including the vascular endothelium, plays a critical role in the different phases of tumor progression leading to invasion. The crosstalk between cancer cells and ECs is therefore crucial for both angiogenesis and cancer invasion. Several studies have reported the importance of endothelial-mediated release of growth factors in cancer chemoattraction. In this regard, TECs have been shown to exert a chemoattractive effect on cancer cells, thus guiding cancer cell metastasis in different cancer types [22,37,38]. In line with this, in directed migration experiments, we demonstrate that compared to normal HMEC, PTEC have a strong effect on PCa cell attraction. Our results indicate that this effect may be due to the expression of TRPC3, which is maintained at high levels in PTEC and downregulated in HMECs. Consistent with this observation, TRPC3 (but not TRPA1 or TRPV2) overexpression in HMEC induced an increase in cell attraction compared to that in control cells. Additionally, PTEC conditioned medium but not HMEC conditioned medium attracted PC3 in directional migration experiments. Moreover, similarly to PTEC conditioned medium, conditioned medium from TRPC3-overexpressing HMECs promoted directional migration of PC3 cells. Although the molecular mechanism underlying EC-driven crosstalk with cancer cells has not been elucidated and requires further studies, we can speculate that TRPC3-mediated Ca^2+^ signaling in TECs may promote the release of growth factors or chemokines, which in turn promote cancer cell migration, as Ca^2+^ signals are known as key players in EC activation in the tumor microenvironment, one of the most relevant steps in tumor progression [4,39].

Finally, we demonstrate that TRPA1 is an important positive modulator of vessel morphogenesis, activating sprouting angiogenesis in vivo and EC migration and tubulogenesis in vitro. We show that endogenous TRPA1 increased wound healing in PTEC and that this effect was inhibited with the downregulation of the channel by siRNA transfection. In line with this, TRPA1 overexpression in HMECs increased cell migration and tubulogenesis in vitro. Most importantly, TRPA1 activation increased angiogenesis in mouse retina, while pharmacological inhibition of TRPA1 caused a decrease in the number of retinal tip cells. Interestingly, the role of Ca^2+^ signals in sprouting angiogenesis has been recently reported in vivo in a zebrafish model [40]. Yokota and coworkers visualized Ca^2+^ dynamics in ECs during sprouting angiogenesis in zebrafish, demonstrating that intracellular Ca^2+^ oscillations occur in ECs exhibiting angiogenic behavior. Dll4/Notch signaling regulates these Ca^2+^ oscillations and is required for the selection of stalk and tip cells during both arterial and venous sprouting [40]. Although whether Dll4/Notch signaling regulates Ca^2+^ stores or Ca^2+^ channels directly or not remains to be delineated, our study suggests TRPA1 as a possible mediator of this pathway, as its activation is clearly involved in vessel sprouting in vivo. In line with these observations we have shown here that TRPA1 agonist induced a strong increase in [Ca^2+^]i only in cells engaged in migration which we could call “tip-like” cells to recall “tip” cell organization in vessel morphogenesis. These data suggest an active role for the channel in sprouting angiogenesis and tip cells activation. Moreover endothelial TRPA1, it has been previously shown to be selectively expressed in cerebral arteries where it plays a role in vasodilation [41]. Indeed, Ca^2+^ influx via endothelial TRPA1 channels elicits vasodilation in cerebral arteries by a mechanism involving Ca^2+^-activated K^+^ channels and inwardly rectifying K^+^ channels in rat myocytes [42]. TRPA1 has also been recently shown to be involved in PCa growth, as its activation by triclosan, a well-established endocrine disruptor, resulted in cytosolic Ca^2+^ influx inducing VEGF secretion by stromal cells, which in turn stimulated epithelial cell proliferation [43]. Moreover, TRPA1 activation by resveratrol induces Ca^2+^ entry, leading to growth factor expression, possibly via the Calcineurin/NFAT pathway, and secretion of HGF and VEGF, which modulate PCa cell growth, migration and resistance to apoptosis [44].

In summary, we performed a complete expression profiling and functional screening that clearly identified four ‘*prostate-associated*’ genes that are deregulated during PCa vascularization, three of which play profound effects on EC biology. In particular, three out of the selected ‘*prostate-associated*’ channels are overexpressed in PCa ECs and have marked effects on the main EC properties including proliferation (TRPV2), TEC-mediated crosstalk with cancer cells (TRPC3) and angiogenesis (TRPA1). The expression profile and functional data could indeed explain the specific transition of PCa to its aggressive phenotype. PTEC have been previously shown to exhibit aggressive phenotypes typical of TECs: indeed, PTEC have a higher migration rate than normal HMECs and are able to form capillary-like structures both in vitro and in xenografts in SCID mice [18]. These results could be of great importance to explain, at least part, the aggressive phenotype previously observed in PTEC relative to that in normal HUVEC and HMEC [18] and provide new putative therapeutic targets.

## 4. Materials and Methods 

### 4.1. Chemicals and Drugs

Naltriben methanesulfonate hydrate (NTB), was resuspended in DMSO to a final concentration of 25 mM and stored at −20°C, according to the manufacturer’s instructions. Allyl isothiocyanate (AITC) was diluted in DMSO to a final concentration of 10 mM and stored at −20 °C, according to the manufacturer’s instructions. L-α-Lysophosphatidylcholine from soybean (LPC) was diluted in ethanol to a final concentration of 20 mM and stored at −20 °C, according to the manufacturer’s instructions. 1-Oleoyl-2-acetyl-sn-glycerol (OAG) was diluted in DMSO to a final concentration of 100 mM and stored at −20 °C, according to the manufacturer’s instructions. All drugs were purchased from Sigma-Aldrich (St Louis, MO, USA). Fura2-AM calcium probe used in calcium imaging experiments was purchased from (Invitrogen Ltd., Waltham, MA, USA) and dissolved in DMSO to a final concentration of 1 mM and stored at −20 °C.

### 4.2. Cell Cultures and Transfection

Normal glomerular ECs (GEC), breast tumor ECs (BTEC), renal tumor ECs (RTEC) were isolated and characterized as described by Bussolati et al. [19,20,45]. Prostate tumor ECs (PTEC) primary cultures were isolated and characterized as previously described [18]. Normal human prostatic microvascular ECs (HPrMEC) were purchased from ScienCell Research Laboratories (Carlsbad, CA, USA). HMECs were obtained from the derma using an anti-CD31 antibody and MACS. HMECs from the derma were immortalized by the infection of primary cultures with a replication-defective adeno-5/SV40 virus as previously described [46,47]. All endothelial cells were cultured in EndoGRO MV-VEGF medium (Merck Millipore, Darmstadt, Germany) containing 5% fetal bovine serum (FBS). All cell cultures were maintained in incubator (37 °C and 5% CO_2_ atmosphere), using Falcon™ plates as supports (about 5000 cells/cm^2^) and were used at passage 3 to 12 (for HMEC, GEC, BTEC, and RTEC) or at passage 1 to 5 (for HPrMEC and PTEC). TECs (BTEC, RTEC and PTEC) were cultured on coating of 1% gelatin. When specified, experiments were performed using DMEM, Dulbecco’s modified eagle medium (Sigma-Aldrich), with 4500 mg/L glucose, 15 mM HEPES and sodium bicarbonate. DMEM was supplemented with 2% L-glutamine, 0.5% gentamicin, and 0, 2, or 10% FBS. HMEC cells were transfected with NucleofectorTM (Amaxa, Gaithersburg, MD, USA) for Human Primary Endothelial Cells with 2 μg of each construct: human pcDNA3TRPA1 [43], human pcDNA3TRPV2 [11], human pcDNA3.1TRPC3 (Addgene, Watertown, MA, USA plasmid #25902), human pcDNA3TRPC6 (already present in our laboratory), mouse pcDNA3.1TRPM7 (kind gift from Prof. Thomas Guderman). Control experiments were performed by transfecting the empty vector. For siRNA-mediated silencing, PTEC were plated in six-well dishes at a concentration of 6 × 10^4^ cells per well (3 wells/condition) the day before oligofection. Oligofection (Oligofectamine, Thermo Fisher Scientific, Waltham, MA, USA) of siRNA duplexes was performed according to manufacturer’s protocol. Briefly, PTEC were transfected twice (at 0 and 24 h) with 200 pmol siLuc (control, siCNTRL), siTRPA1 or siTRPV2. siRNA against Luciferase (siLuc; Eurogentec, Seraing, Belgium) was used for control silencing. 24 or 48 h after the second oligofection, PTEC were lysed or tested in functional assays. The siTRPA1 sequence is 5′-GGUGGGAUGUUAUUCCAUA(dTdT)-3′ [43], and the siTRPV2 sequence is 5′-UAAGAGUCAACCUCAACUAdTdT (dTdT)-3′ [32].

### 4.3. Generation of Immortalized PTEC

Primary prostate-derived TEC (PTEC) were infected (at passage 2) with a retrovirus containing a pBABE-puro-hTERT plasmid (Addgene plasmid #1771) [48] and selected using the antibiotic resistance (1 μg/mL puromycin, Gibco, Thermo Fisher Scientific, Waltham, MA, USA) for two weeks. hTERT mRNA was always upregulated in hTERT PTEC as compared with wild type PTEC, as shown in Appendix A.

### 4.4. Flow Cytometry

hTERT PTEC were detached from plates with a non-enzymatic cell dissociation solution (Sigma-Aldrich), washed and stained (30 min at 4 °C) with the following fluorescein isothiocyanate (FITC)-, phycoerythrin (PE)-, or allophycocyanin (APC)-conjugated antibodies: CD31 (BD Bioscience, Franklin Lakes, NJ, USA), CD105 (from MiltenyiBiotec, Bergisch Gladbach, Germany), VEGFR1 (R&D Systems, Minneapolis, MN, USA). Isotypes (all from MiltenyiBiotec, Bergisch Gladbach, Germany) were used as negative controls. Cells were subjected to cytofluorimetric analysis (FACScan Becton Dickinson, Franklin Lakes, NJ, USA) every other passage.

### 4.5. In Vivo Angiogenesis

Wild type outbred OF1 mice were purchased from Charles River (Wilmington, MA, USA). Sub-retinal injections were performed in three-day-old anesthetized mice by injecting 0.5 µL DMSO (control), 500 µM AITC or 500 µM HC030031. Six-day-old mice were sacrificed by decapitation according to the CNRS recommendation and eyes were enucleated. Retinal cups were dissected under binoculars and briefly fixated using 4% PFA in PBS before whole mount immunostaining. Flat mount retinas were incubated overnight at 4 °C with the rabbit anti-mouse type IV collagen (ab6586, 1 mg/mL, 1:500, Abcam, Cambridge, UK). After washes, retinas were then incubated with the fluorescent secondary antibody donkey anti-rabbit A594 (Invitrogen, 2 mg/mL, 1:500). Retinas were finally washed and mounted in Mowiol before imaging with an AxioImager Z1-Apotome (Carl Zeiss, Marly le Roi, France). All pups derived from a litter were used in one experiment. Experiments were performed two times with six retinas per treatment. The distances between the established retina vascular plexus and the extended vascular sprouts were measured for each condition. VM was authorized to perform experimentation on animals (#59-35066) and experimental procedure has been deposited to Ministère de l’enseignement supérieur et de la Recherche (French government research department), as per regulations.

### 4.6. Isolation of CD31 Positive Cells from Retinas

Retinas were harvested from six-day-old mouse pups previously sacrificed by decapitation. Retinas were digested with type I collagenase (2 mg/mL, Life Technologies) and DNAse I (10 µg/mL, Roche Life Science, New York, NY, USA) in DMEM for 40 min. Cell suspensions were then incubated with 25 µL magnetic beads (Dynabeads Sheep anti-Rat IgG, Life Technologies) previously coated with 1.5 µg anti-CD31 antibody (rat anti-mouse CD31, Becton Dickinson). CD31+ cells were isolated following manufacturer’s instructions. Isolated cells were centrifuged and suspended in Trizol for total RNA isolation and RT-qPCR analysis.

### 4.7. Patients and Tissues

Formalin-fixed paraffin-embedded tissue samples containing both normal and tumor areas were obtained from 10 patients treated by radical prostatectomy for prostate cancer. Patients were aged 62 to 70 years old, with PCa classified as ISUP group 2 in 6 cases and group 3 in 4 cases. Serial 3 µm sections of the paraffin blocks were used for immunohistochemistry. Written informed consents were obtained from patients in accordance with the requirements of the medical ethic committee, Comité de Protection des Personnes of Tours Universitary Hospital of the 18th of February (DC-2014-2045).

### 4.8. Immunohistochemistry

Slides were deparaffinized, rehydrated, and heated in citrate buffer pH 6 for antigenic retrieval. After blocking for endogenous peroxidase with 3% hydrogen peroxide, the following antibodies were incubated: TRPA1 (#ACC-037 1:500, 1 h, Alomone Labs, Jerusalem, Israel), TRPV2 (SAB1101376 Sigma, 1:1000, 1 h), TRPC3 (#ACC-016 Alomone, 1:500, 1 h), TRPC6 (#ACC-017 Alomone, 1:300, 1 h). Immunohistochemistry was performed using the streptavidin-biotin-peroxidase method with diaminobenzidine as the chromogen (Kit LSAB, Dakocytomotion, Glostrup, Denmark). Slides were finally counterstained with haematoxylin. Negative controls were obtained after omission of the primary antibody or incubation with an irrelevant antibody. For cryo-sections, retinas were fixed overnight with 4% PFA, rinsed in PBS and then incubated with 30% sucrose before OCT embedding. Retina sections were blocked and permeabilized in PBS 0.25% triton, 5% FBS, 1% BSA for 2 h and incubated overnight at 4 °C with the rabbit anti-TRPA1 antibody (#ACC-037 Alomone, 1:100) and the rat anti-CD31 (Becton Dickinson, 1/100). After washes, sections were incubated with secondary antibody donkey anti-rabbit A488 and donkey anti-Rat A594 for 2 h at room temperature. Sections were then washed and stained with DAPI before mounting in Mowiol. Sections were imaged with a Carl Zeiss AxioImager Z1-Apotome.

### 4.9. Total RNA Extraction and Reverse Transcription

Total RNAs were extracted from cultured cells by Nucleospin RNA II kit (Clontech Laboratories, Mountain View, CA, USA) according to manufacturer’s protocol, and subjected to reverse transcription as previously described [49].

### 4.10. Quantitative Real-Time PCR

Real-time qPCR of cDNA was done using qPCR SsoFast™ EvaGreen^®^ Supermix (Bio-Rad, Hercules, CA, USA) on the CFX96 Real-Time PCR Detection System (Bio-Rad). The primers for *trp* channels were designed on the functional region of the channels using Primer3 software (PREMIER Biosoft, Palo Alto, CA, USA) and efficiency was validated on *trp*-coding plasmids and cell lines expressing the channels (Table 1).

The sequences for the *actin* and *hprt* primers are: 5′-CAGCTTCCGGGAAACCAAAGTC-3′ and 5’-AATTAAGCCGCAGGCTCCACTC-3′ for *18s*; 5′-GGCGTCGTGATTAGTGATGAT-3′ and 5′-CGAGCAAGACGTTCAGTCCT-3′ for *hprt.* For the qPCR on cDNA from mice retina the following primers were used: 5′- GCAGGTGGAACTTCATACCAACT-3′ and 5′-CACTTTGCGTAAGTA CCAGAGTGG-3’ for *trpa1*; 5′-CTGCAGGCATCGGCAAA-3′ and 5′-GCATTTCGCACACCTGGAT-3’ for *CD31*; 5’-GCCATGGATGACGATATCGCTG-3′ and 5′-GCCATGGATGACGATATCGCTG-3′ for *actin*. *18s* and *hprt* (hypoxanthine-guanine phosphoribosyltransferase) were used as internal controls to normalize variations in RNA extraction and RT efficiency. In order to quantify relative gene expression (Figure 1d, Appendix A), the delta-delta Cycle threshold (ΔΔ*C_T_*) method was used, calculating the difference between the normalized expression values of tumor and normal samples: $2^{-\Delta \Delta C_{T}}$, where $\Delta \Delta \mathrm{Ct}=({\Delta C}_{T\;target\;gene,\;\mathrm{tumor}}\;–{\;\Delta C}_{T\;ref\;gene,\;\mathrm{tumor}})-\left({\mathrm{mean}\;\Delta C}_{T\;target\;gene,\;\mathrm{normal}}\;–{\;\mathrm{mean}\;\Delta C}_{T\;ref\;gene,\;\mathrm{normal}}\right)$. In dispersion graphics (Figure 1e), values are expressed as mean $Log_{10}\;\left(\frac{C_{T,\;target\;gene}}{C_{T,\;18s}}\right)$ of at least three independent experiments. All screened ECs did not show differences in the expression of the two internal controls used for the screening, *18s* and *hprt* (Appendix A), showing that there were no significant variations in the amount of input cDNA that could interfere with the differential profiling. Prior to the definition of the TEC differential profile, we had to assess the correct healthy counterpart to be used for PTEC. We therefore plotted the mRNA expression of two normal EC cell lines, namely dermal HMEC and prostatic HPrMEC, of the different *trp* channels (expressed as the Logarithm of the corrected qPCR threshold cycle (*C_T_*) values). As shown in Appendix A, the values are very poorly correlated: most of the values of HMEC are stable in contrast to the values of HPrMEC, showing the necessity of a prostate specific EC model as healthy counterpart for our studies. In light of these results, we decided to use HPrMEC as the healthy counterpart of PTEC.

### 4.11. Western Blot

Conditions for SDS–PAGE and western blotting were as previously described [8]. Polyvinylidene fluoride membranes were properly blocked in 5% bovin sieric albumin (BSA) in TNT buffer (0.1 M Tris-Cl pH 7.5, 150 mM NaCl, 0.1% Tween-20) for 30 s and then incubated over night with anti-TRPA1 (#ACC-037, Alomone, 1:200), anti-TRPC3 (#ACC-016, Alomone, 1:500), anti-TRPC6 (#ACC-017, Alomone, 1:200), or anti-TRPV2 (SAB1101376, Sigma, 1:500), anti-β-actin (A5316, Sigma, 1:1000) primary antibodies, following manufacturer’s instructions. The membrane was then washed using TNT containing 0.1% Tween 20 and incubated with the appropriate HRP-conjugated secondary antibodies (SantaCruz, Dallas, TX, USA). Membranes were treated with either Femto or Dura enhanced chemiluminescence (ECL) reagents (ThermoFisher, Thermo Fisher Scientific, Waltham, MA, USA) for 1 or 5 min respectively, and exposed by Amersham Imager 600 (GE Healthcare, Little Chalfont, UK). To quantify the differences in protein expression, the ratio between TRP channels and actin expression was evaluated using Fiji, ImageJ software (https://imagej.net/Fiji). Whole scans of the immunoblots were shown in Appendix A.

### 4.12. Calcium Imaging

Cells were seeded on gelatin-coated glass coverslips at a density of 5000 cells/cm^2^ at 24 h before the experiments. Cells were starved in DMEM 5% FBS for at least 2 h before the experiments. Cells were next loaded (45 s at 37 °C) with 2 μM Fura-2 AM, for ratiometric cytosolic calcium concentration ([Ca^2+^]_i_) measurements. During experiments, cells were maintained in standard extracellular solution of the following composition: 154 mM NaCl, 4 mM KCl, 2 mM CaCl_2_, 1 mM MgCl_2_, 5 mM HEPES, 5.5 mM glucose. Solution pH was adjusted to 7.35 using NaOH. 4 h before experiments cells were starved in DMEM 2% FBS. Fluorescence measurements were made using a Polychrome V spectrofluorimeter (TILL Photonics, Munich, Germany) attached to an Olympus ×51 microscope (Olympus, Tokyo, Japan) and Metafluor Imaging System (Molecular Devices, Sunnyvale, CA, USA). [Ca^2+^]_i_ was measured using ratiometric probe Fura-2-AM and quantified according to Fiorio Pla et al. [50].

### 4.13. Cell Viability and Proliferation Assay

Cells were transfected by NucleofectorTM (Amaxa, Gaithersburg, MA, USA) with 2 μg of each construct and plated (1600 cells/well) on 96-well plates in EndoGRO MV-VEGF medium. 20 h after transfection, cells were starved in DMEM containing 2% FBS for 4 h. After starvation, EndoGRO MV-VEGF and DMEM 2% FBS with or without agents to be tested were added (12 wells/condition) to cutured cells. EndoGRO MV-VEGF was used as positive control, whereas 2% FBS served as negative control. 24, 48 and 72 h after treatments, cells were washed with PBS and colored using the CellTiter 96 AQueous Non-Radioactive cell proliferation assay (Promega, Madison, WI, USA). Assays were performed by adding 40 μL of MTS-containing solution to each culture well and absorbance was recorded at 495 nm in a microplate reader (Dinnex Technologies MR422, Thermo Labsystems, Philadelphia, PA, USA) after 2 and 4 h of incubation.

For siRNA-mediated silencing on hTERT PTEC, Oligofection (Oligofectamine, Life Science) of siRNA duplexes was performed twice (at 0 and 24 h) with 200 pmol siLuc (control, siCNTRL) or siTRPV2 (see also “Cell cultures and transfection” section). 6 h after the second pulse hTERT PTEC were plated (2500 cells/well) on 96-well plates in EndoGRO MV-VEGF medium. Proliferation was evaluated using Cell Proliferation ELISA, BrdU (colorimetric, Roche) following manufacturer instruction. Briefly, 48 h after the first pulse, cells were labelled with BrDU 10 µL/well overnight. ELISA assay was performed by adding Add 100 µL/ well anti-BrdU-POD working solution for 90 min and subsequently adding 100 µL/well Substrate solution until color development is sufficient for photometric detection (5–30 min). Absorbance was measureed the absorbance of the samples in the microplate reader at 370 nm (reference wavelength: approx. 492 nm).

### 4.14. Scratch-Wound Healing Assay

Cells were transfected by NucleofectorTM with 2 μg of each construct, as previously described, and plated (10 × 10^4^ cells/well) using EndoGRO MV-VEGF on 24-well culture plates coated with 1% gelatin. 20 h after transfection, cell monolayers were starved for 4 h in DMEM 0% FBS. Motility assay was performed by generating a wound in the confluent cellular monolayers by means of a P10-pipette tip. Floating cells were removed by two washes in PBS solution, and monolayers were treated with test conditions (in duplicate). Complete EndoGRO 5% FBS was used as positive control, whereas DMEM 0% FBS served as negative control. Experiments were performed using a Nikon Eclipse Ti (Nikon Corporation, Tokyo, Japan) inverted microscope equipped with a A.S.I. MS-2000 stage and a OkoLab incubator (to keep cells at 37 °C and 5% CO_2_). Images were acquired at 2 h time intervals for 5 time points, using a Nikon Plan 4×/0.10 objective and a CCD camera. Within two subsequent time points, MetaMorph software (Molecular Devices) was used to measure the distance covered by cells to close the “wound area” (four field measurements for each image, at least 10 fields for each condition analyzed in each independent experiment) and to calculate migration rate (%).

### 4.15. Random Migration Assay

HMEC were transfected by NucleofectorTM with 2 μg of each construct. Cells were plated (1.5 × 10^4^ cells/well), using EndoGRO MV-VEGF for HMEC on 24-well culture plates coated with 1% gelatin. 20 h after transfection, cells were starved for 4 h in DMEM 0%. EndoGRO 5% FBS was used as positive control, whereas DMEM 0% FBS was the negative control for HMEC. Experiments were performed using the same set-up described for the wound healing assays but using a Nikon Plan 10×/0.10 objective. Images were acquired for 10 h every 10 min using MetaMorph software. Image stacks were analyzed with ImageJ software and at least 500 cells/condition were tracked. Migration rate (μm/s) is obtained by measuring the distance covered by cells between two subsequent time points after conversion of pixels to micrometers. Directional migration index, or cell persistence, was calculated as the Euclidean distance from the initial position of each cell to the final point of its path (i.e., straight line from initial to final positions) divided by the total path length (sum of the distances covered between each acquisition). The value obtained was then normalized to the total duration of the cell tracking and then multiplied by the square root of the cell tracking duration, as previously described [49].

### 4.16. Cell Attraction Assay

HMEC (10 × 10^4^ cells/well) and PTEC (5 × 10^4^ cells/well) were plated in 24-well plates (2 wells/condition) using EndoGRO MV-VEGF medium. After 24 h, Transwell^®^ permeable supports (6.5 mm inserts with an 8 µm pore polycarbonate membrane) were equilibrated for 20 s at 37 °C using RPMI 0% FBS medium. The equilibrated Transwell^®^ inserts were then placed over the wells containing the previously plated HMEC and PTEC. PC3 cells (5 × 10^4^ cells/insert) were seeded in the Transwell^®^ inserts using RPMI 0% FBS and incubated for 24 h. Transwell^®^ inserts were then washed in PBS twice and fixed in methanol for 30’. PC3 cells were then colored with 0.5% Crystal Violet in methanol for 20 s at room temperature and after washed in PBS. PC3 cells that did not migrate thought the membrane pores were removed from the upper side of the membrane using a cotton bud. PC3 cells that migrated thought the membrane pores were then counted using an Eclipse Ti-E Nikon microscope with a 10×/0.25 NA Plan objective.

### 4.17. 2D Chemotaxis Assay

6 µL of PC3 cell suspension (3 × 10^6^ cells/mL) were plated in the observation area (3 observation areas/chamber) of a µ-Slide Chemotaxis chamber (ibidi GmbH, Gräfelfing, Germany) using RPMI 10% FBS medium and incubated to allow cell attachment. After 6 h, 65 µL of control medium (EndoGRO basal medium) were added to the right reservoirs of the chamber (3 right reservoirs/chamber). 65 µL of control basal medium or PTEC/HMEC/TRPC3-overexpressing HMEC conditioned medium were added to the left reservoirs of the chamber (3 left reservoirs/chamber). Experiments were performed using the same set-up described for the wound healing assays with a Nikon Plan 10×/0.10 objective. Images of the observations chambers were acquired for 10 h every 10 min using MetaMorph software. Image stacks were analyzed with ImageJ software.

### 4.18. In Vitro Tubulogenesis

*In Vitro* formation of capillary-like structures was studied on growth factor-reduced Matrigel (Corning, Corning, NY, USA) for HMEC and a solution of 40% growth factor-reduced Matrigel and 60% Collagen-I 4 mg/mL gel obtained by mixing 1 M HEPES, 37 g/L NaHCO3, and 5 mg/mL collagen-I in a 1:1:8 ratio (Cultrex 3D Culture Matrix Rat Collagen I, Amsbio) for PTEC-hTERT. HMEC were transfected by NucleofectorTM with 2 μg of each construct, as previously described. 24 h after transfection, cells were seeded (3.5 × 10^4^ cells/ well) onto Matrigel-coated 24-well plates in growth medium containing treatments (in duplicate). For siRNA-mediated silencing on PTEC-hTERT, Oligofection (Oligofectamine, Life Science) of siRNA duplexes was performed twice (at 0 and 24 h) with 200 pmol siLuc (control, siCNTRL) or siTRPA1 (see also “Cell cultures and transfection” section). EndoGRO 5% FBS was used as positive control and DMEM 10% FBS as negative control. Cell organization onto Matrigel was periodically observed with the same set-up as the one used for wound healing and migration assays using a Nikon Plan 10×/0.10 objective. Images were acquired at 2 h time intervals (10 time points) using MetaMorph software. In order to quantify tubulogenesis images, we used the Angiogenesis Analyzer plugin of the ImageJ software, developed by Gilles Carpentier (ImageJ contribution: Angiogenesis Analyzer. ImageJ News, 5 October 2012). Angiogenesis Analyzer allows the analysis of cellular networks by the detection of different characteristics and constitutive elements of tubules. In order to measure differences in cell ability to from pseudo-capillary structures in vitro, we quantified the number of master junctions and the total master segments length. Master junctions are defined as portions of tree delimited by two junctions, none exclusively implicated with one branch. The total master segment length is the sum of the length of the detected master segments in the analyzed area, where master segments are the elements delimited by two master junctions.

### 4.19. Statistical Analysis

Statistical analysis was performed using R 3.4.2 software and Kaleidagraph Software (Synergy Software, Reading PA, USA). Statistical significance between populations was determined by Student’s *t*-test (for normal populations) or non-parametric Wilcoxon-Mann-Whithney test. Differences with *p*-values < 0.05 were considered statistically significant. Heatmap: A heatmap was generated to represent and visualize the relative expression pattern of TRP genes. Figure 1a was produced with Complex Heatmap package 1.14.0. (https://bioconductor.org/packages/release/bioc/html/ComplexHeatmap.html). Euclidean distance and Ward’s aggregation criterion were used to perform the hierarchical classification of the channels. The color key displayed in heatmap was prepared by putting the color breaks to log2 fold change values as follows: red, −4 to −2; black, −2 to +2 and green, 2 to 4. Normalization of relative expressions patterns: all experiments, except the ones involving TRPP2, were carried with at least two technical and three biological replicates for each experimental condition. For TRPP2, only two biological replicates were used. All initial values (ΔΔC_T_) were log_2_ transformed in order to stabilize the variance and directly observe the log_2_ fold changes (log2FC). Values of technical replicates were averaged before the log_2_ transformations. Relative expression pattern was obtained by subtraction of the healthy counterpart to the log_2_-averages of each biological sample. Differential analysis: Differential analysis of ΔΔC_T_ between PTEC1-3 samples and HPrMEC samples was performed with non-parametric Wilcoxon tests for unpaired data, without averaging technical replicates. In order to take into account the multiple testing issue, raw *p*-values were adjusted using the Benjamini-Hochberg method [51], which controls the false discovery rate (FDR). Differences with adjusted *p*-values < 0.05 were considered statistically significant except for Figure 1b for which significance was at 0.1 (Figure 1b).

## 5. Conclusions

We performed a complete expression profiling and functional screening that clearly identified four ‘*prostate-associated*’ genes that are deregulated during PCa vascularization, three of which have profound effects on EC biology. In particular, three out of the selected ‘*prostate-associated*’ channels are overexpressed in PCa ECs and have marked effects on the main EC properties including proliferation (TRPV2), TEC-mediated crosstalk with cancer cells (TRPC3) and angiogenesis (TRPA1). The expression profile and functional data could indeed explain the specific transition of PCa to its aggressive phenotype. PTEC have been previously shown to exhibit aggressive phenotypes typical of TECs: indeed, PTEC have a higher migration rate than normal HMECs and are able to form capillary-like structures both in vitro and in xenografts in SCID mice [18]. These results could be of great importance to explain, at least part, the aggressive phenotype previously observed in PTEC relative to that in normal HUVEC and HMEC [18] and provide new putative therapeutic targets.

## 6. Patents

The authors disclose the Report of Invention: “Set de oliginucleotides de qPCR pour l’établissement de profil d’expression du domaine fonctionnel des 26 canaux de type TRP” Reference # DSO2018005437), Patent Pending (D.G., M.B., N.P.) and thus the exact primer sequences and positions under declaration of invention cannot be stated.

## Figures and Tables

**Figure 1 cancers-11-00956-f001:**
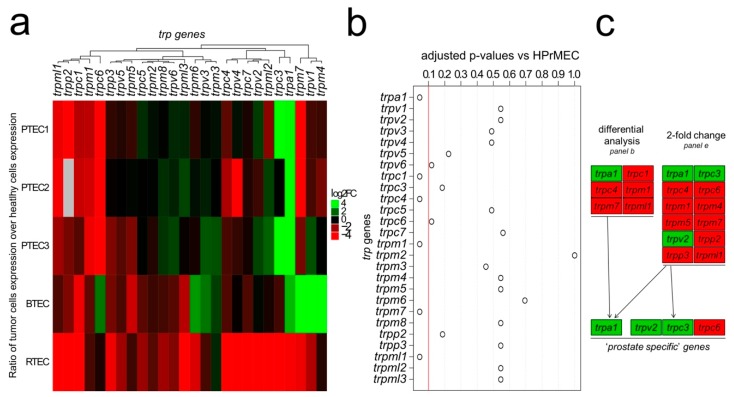
Identification of the *trp* expression signature in PTEC. (**a**) Heatmap showing the relative mRNA expression of the *trp* genes in TECs of different origins. TEC expression normalized based on normal endothelial counterparts: PTEC1-3 expression is normalized against HPrMEC expression, while BTEC and RTEC expression is normalized against HMEC and GEC expression, respectively. (**b**) Differential analysis of PTEC1-3 and HPrMEC highlighted six genes (i.e., *trpa1*, *trpc1*, *trpc4*, *trpm1*, *trpm7*, and *trpml1*) that are differentially deregulated (adjusted *p*-value < 0.1, red line), while the other genes did not show significant deregulation. (**c**) Differential analysis identified six genes that are differentially deregulated (panels **a**,**b**), and twelve genes displayed at least 2-fold change in their expression in at least two patient lines (panel **e**). Among these, four channels (*trpa1*, *trpv2*, *trpc3* and *trpc6*) were selected as ‘*prostate-associated*’. (**d**) Real-time qPCR analysis of mRNA expression shows that *trpa1*, *trpv2*, *trpc3* and *trpc6* are deregulated in PTEC1-3 relative to those in normal HPrMEC, and the deregulation is observed only in the prostatic tissue but not in the other normal ECs or TECs tested. Values are expressed as the mean ΔΔCT values ± SEM of at least three independent experiments. *: *p*-value < 0.05. (**e**) Scatter plots showing the expression of *trp* channels in PTEC1 (left panel), PTEC2 (central panel) and PTEC3 (right panel) relative to that in HPrMEC, as determined by real-time qPCR. Gray dots indicate genes with less than 2-fold change in PTEC relative to those in HPrMEC. Blue dots indicate genes with more than 2-fold change in at least one PTEC line relative to those in HPrMEC. Red dots showing the four ‘*prostate-associated*’ genes (i.e., *trpa1*, *trpv2*, *trpc3*, and *trpc6*), with more than 2-fold change in at least 2 PTEC lines out of three. Correlation values for *trp* channels are expressed as Log (C_(T,gene x)/C_(T,18s)) in primary PTEC (*x*-axis) and normal HPrMECs (*y*-axis).

**Figure 2 cancers-11-00956-f002:**
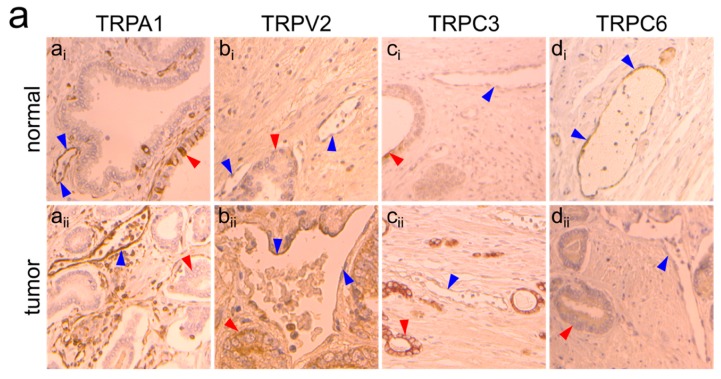
Protein expression of ‘*prostate-associated*’ genes in PTEC and HPrMEC. (**a**) (panel **a_i_**, ×20) TRPA1 expression in normal areas: focal positive staining in epithelial cells (red arrows) and diffuse staining in ECs (blue arrows). (panel **a_ii_**, ×20) In tumor areas, TRPA1 expression is observed in all cases (*n* = 10/10) in ECs (blue arrows), with negative staining in tumor cells (red arrows). (panel **b_i_**, ×40) In histologically normal areas, TRPV2 expression is not observed in ECs (*n* = 0/10) (blue arrows) or epithelial cells (red arrows). (panel **b_ii_**, ×40) Positive endothelial TRPV2 expression was observed in 7 of 10 cases in the tumor areas (blue arrows), with positive staining in cancer cells in 4 cases (red arrow). TRPC3 expression: (panel **c_i_**, ×40) all normal tissues showed negative staining (*n* = 10/10) in ECs (blue arrows) with focal positive staining in epithelial cells (red arrow). (panel **c_ii_**, ×10) Only one tumor (*n* = 1/10) showed faint positive TRPC3 staining on ECs (blue arrow), whereas cancer cells were positive (red arrows) in all cases. (panel **d_i_**, ×20) TRPC6 expression was observed in ECs in histologically normal tissues (*n* = 6/10), (panel **d_ii_**, ×40) whereas TRPC6 expression was absent in cancer cells in 7 out of 10 patient samples (red arrows). (**b**) Summary of endothelial and epithelial expression of the four ‘*prostate-associated*’ candidates in both normal and tumor areas for the ten patient samples used for immunohistochemical analysis. The patients were 62 to 70 years old, with PCa classified as ISUP group 2 in six cases and group 3 in four cases. Ca^2+^- imaging traces in response to TRPA1, TRPV2 and TRPC3 agonists respectively (**c**) 20 µM AITC), (**d**) 10 µM LPC and (**e**) 50 µM OAG in PTEC1 (black), PTEC2 (blue) and PTEC3 (red) but not in HPrMEC (green).Traces represents mean ± SEM of cells in the recorded field of one representative experiment. Lower panel: histogram showing peak amplitude of TRP agonist-mediated Ca^2+^ responses (mean ± 95% CI of different cells in the field from at least 3 independent experiments). Statistical significance and ****: *p*-value < 0.0001 (Kruskal-Wallis test).

**Figure 3 cancers-11-00956-f003:**
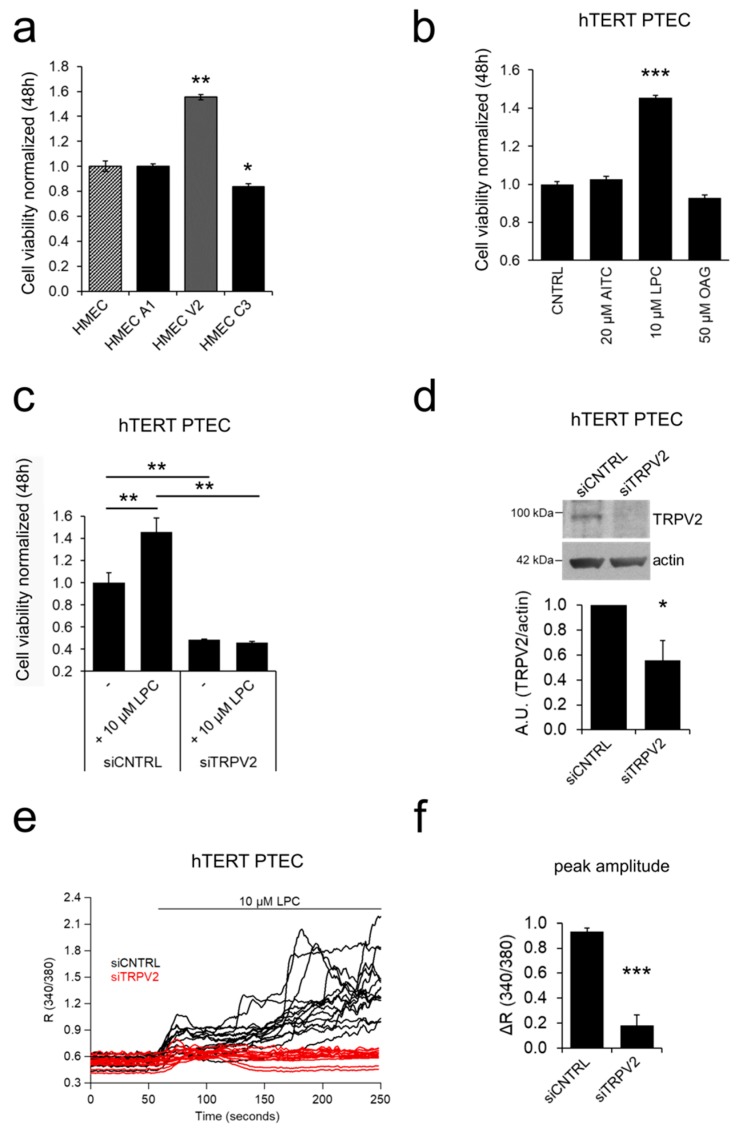
TRPV2 modulates EC viability. (**a**) The ‘*prostate-associated*’ channels were overexpressed in HMEC to test their effect on cell viability. Of all channels, only TRPV2-overexpression positively modulated proliferation in HMEC, 48 hours after transfection. Statistical significance *: *p*-value < 0.05 and **: *p*-value < 0.005 vs HMEC. (**b**) Treatment with the ‘*prostate-associated*’ channel agonists shows that 10 µM LPC induces an increase in hTERT PTEC viability 48 h after treatment. Agonists were added to the basal growth medium. Data represent the mean ± SEM of a minimum of three independent experiments. Statistical significance ***: *p*-value < 0.005 vs. control basal medium. (**c**) Silencing of TRPV2 expression significantly reduced hTERT PTEC viability, under both basal conditions and LPC stimulation and reverted the LPC-induced increase in PTEC viability observed in control cells. Statistical significance **: *p*-value < 0.005. (**d**) TRPV2 downregulation 48 h after siRNA transfection was verified by western blotting in hTERT PTECs. Histogram showing the quantification of TRPV2 expression relative to actin represented as the mean ± SEM of a minimum of three independent experiments. (**e**) Downregulation of TRPV2 expression was also studied in Ca^2+^ imaging experiments 48 h after siRNA transfection by stimulation with 10 µM LPC in control and TRPV2-silenced hTERT PTEC. Each trace represents the ratio (340/380 nm) of a single cell in the field in one representative experiment.

**Figure 4 cancers-11-00956-f004:**
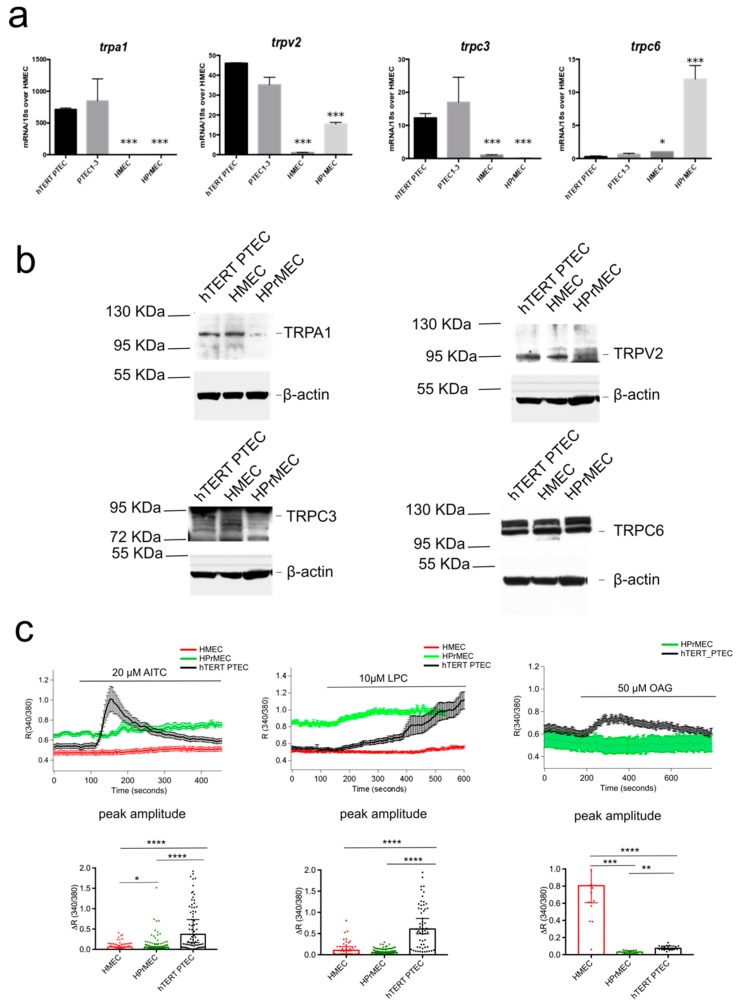
hTERT PTEC validation as cellular model for TEC studies. (**a**) Real-time qPCR analysis validating equal amount of mRNA expression if *trpa1*, *trpv2*, *trpc3* and *trpc6* in hTERT PTEC compared to PTEC1-3, as well as deregulated expression compared to HMEC and HPrMEC. Values are expressed as the mean ΔΔCT values ± SEM of three independent experiments. (**b**) Representative immunoblots blots (*n* = 3) showing similar TRPA1, TRPV2 and TRPC3, and TRPC6 expression in hTERT PTEC compared to PTEC1, 2 and 3. Β-actin was used as a loading control. (**c**) Ca^2+^- imaging traces in response to TRPA1, TRPV2 and TRPC3 agonists respectively 20 µM AITC), 10 µM LPC and 50 µM OAG in HMEC (red), HPrMEC (green) and hTERT PTEC (black). Traces represents mean ± SEM of all cells in the recorded field of one representative experiment. Lower panel: histogram showing peak amplitude of TRP agonist-mediated Ca^2+^ responses (median ± 95% CI of different cells in the field from at least three independent experiments). Statistical significance *: *p*-value < 0.05, **: *p*-value < 0.005 and ****: *p*-value < 0.0001 (Kruskal-Wallis test.).

**Figure 5 cancers-11-00956-f005:**
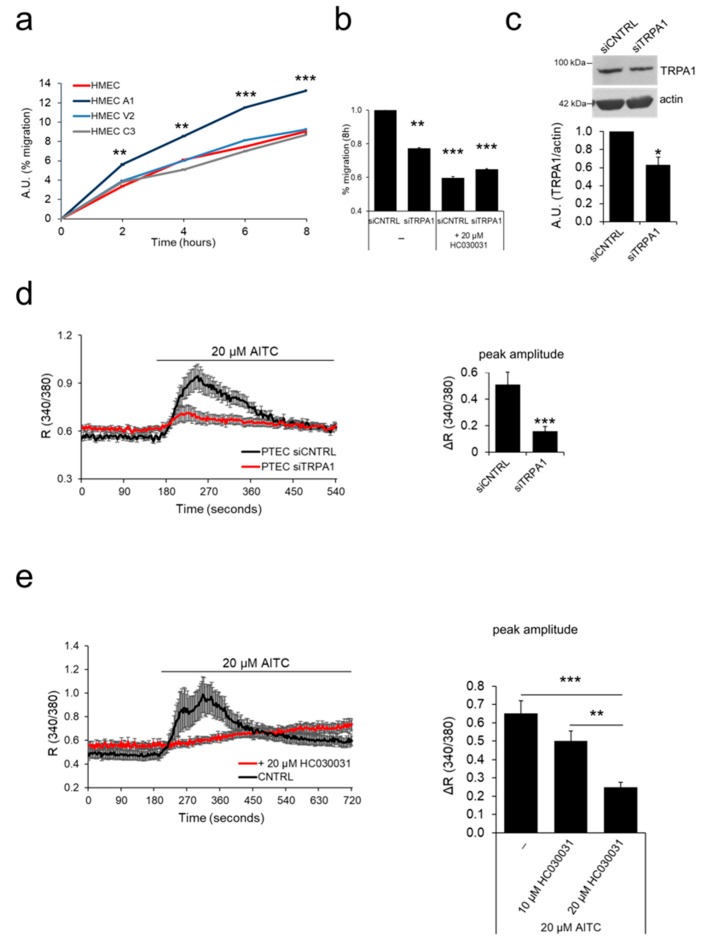
TRPA1 increases EC migration. (**a**) The ‘prostate-associated’ channels were overexpressed in HMECs to test their effect on cell migration. Expression of TRPA1 increased the migration of HMEC relative to control cells in basal medium, as observed by wound healing experiments. Lines show the mean value ± SEM of one out of three independent experiments (24 h after transfection). Statistical significance **: *p*-value < 0.005 and ***: *p*-value < 0.0005 vs. HMEC. (**b**) Histogram showing the % migration in wound healing experiments of PTEC transfected with control siRNA (siCNTRL) or siRNA against TRPV2 (siTRPV2), under both basal conditions and 8 h after treatment with TRPA1 inhibitor HC030031. Statistical significance **: *p*-value < 0.005 and ***: *p*-value < 0.0005 vs. HMECs. (**c**) TRPA1 downregulation 48 h after siRNA transfection was verified by western blotting. Histogram shows quantification of TRPA1 expression relative to actin represented as the mean + SEM of a minimum of three independent experiments. *: *p*-value < 0.05. (**d**) Downregulation of TRPA1 expression was verified 72 h after siRNA transfection in Ca^2+^ imaging experiments by stimulation with 20 µM AITC in control PTEC (siCNTRL, black trace) and in PTECs transfected with siRNA against TRPA1 (siTRPA1, red trace). (**e**) TRPA1 activity is inhibited when PTEC are treated with a TRPA1 channel inhibitor (20 µM HC030031) based on Ca^2+^ imaging experiments. For experiments in **c**, **d** and **e** traces show the mean value ± SEM of all cells in the recorded field of one representative experiment (left panel); histograms represent the quantification of the peak amplitude after treatment with AITC (**c**,**d**) or different doses of the channel inhibitor (10 or 20 µM HC030031) in Ca^2+^ imaging experiments (**e**, right panel).

**Figure 6 cancers-11-00956-f006:**
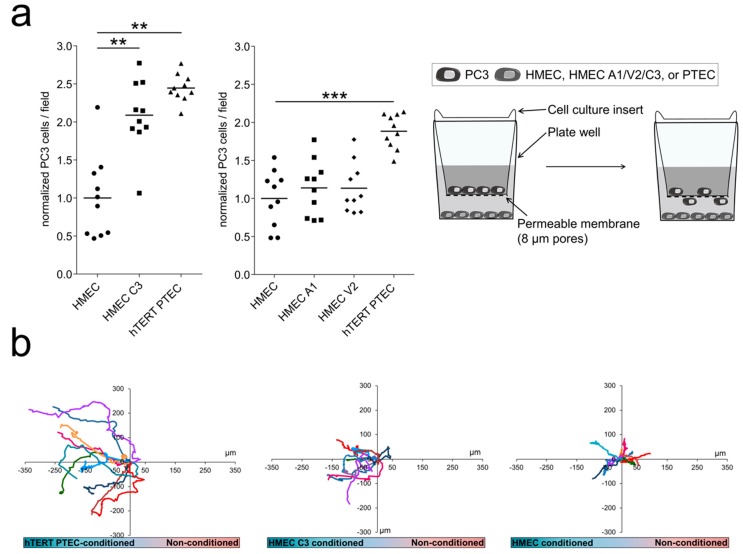
TRPC3 has a role in PCa cell attraction. (**a**) PC3 cells were used to study the effect of TRPA1, TRPV2 and TRPC3 on cancer cell attraction. Control HMEC, channel-overexpressing HMEC or hTERT PTEC were plated in multiwell plates, and PC3 cells were plated in the upper chamber of 8 µm-pore culture inserts. hTERT PTEC or TRPC3-overexpressing HMEC (HMEC C3) increased PC3 cells migration through the membrane, quantified 24 h after plating. Each dot indicates the cell number in a single field normalized to PC3 cell count; results are presented as normalized values against co-cultured HMEC, while the continuous line represents the median value of normalized cells/field for each condition (*n* = 10). The graph shows one representative experiment out of three. Statistical significance: ** *p*-value < 0.005; *** *p*-value < 0.0005. Right panel shows the schematic representation of the migration assays used to study PC3 cell migration in co-culture with either hTERT PTEC, TRPA1/V2/C3-overexpressing HMEC, or control HMEC. (**b**) Chemotaxis chambers were used to study hTERT PTEC-directed migration in the presence of hTERT PTEC-, TRPC3-overexpressing HMEC-, HMEC- or non-conditioned (non-cond) media. Compared to HMEC conditioned medium, PTEC conditioned medium (24-h conditioning) increased PC3 cell speed (**c**, left panel), while compared to control HMEC conditioned medium, the conditioned medium from TRPC3-overexpressing HMECs significantly promoted PC3 migration via directional persistence index and not on the speed (**c**, central and right panels). Statistical significance: * *p*-value < 0.05, ** *p*-value < 0.005 and *** *p*-value < 0.0005.

**Figure 7 cancers-11-00956-f007:**
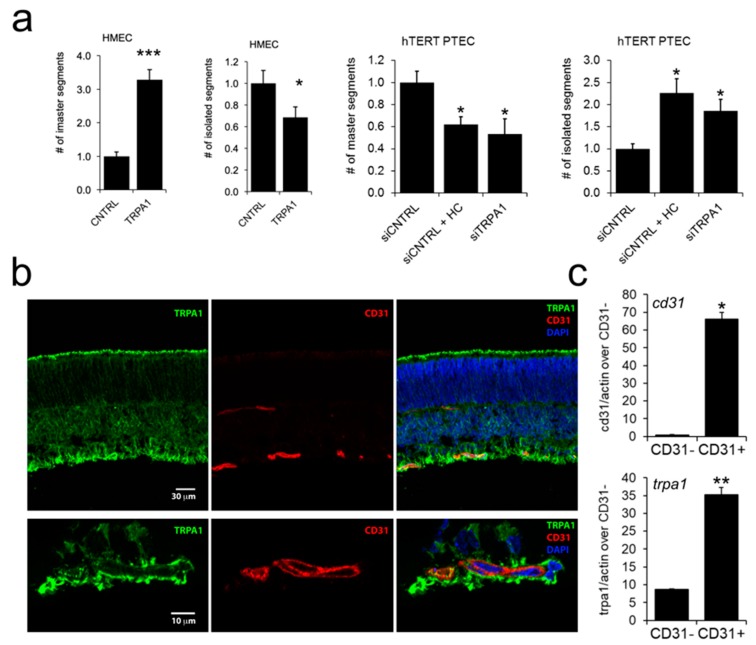
TRPA1 promotes in vitro tubulogenesis and in vivo angiogenesis. (**a**) left panels: TRPA1-overexpressing HMEC display increased in vitro tubulogenesis in Matrigel compared to control cells; right panels: TRPA1-downregulating hTERT PTEC display decreased in vitro tubulogenesis in Matrigel/collagenI substrate compared to control cells; bar graphs show the quantification of number of master segment and the number of isolated segments of control HMECs and TRPA1-overexpressing HMECs. Three independent experiments were performed 24 h after overexpression or after II oligofection pulse. (**b**) Endogenous TRPA1 is diffusely expressed in postnatal mouse retina but clearly co-localizes with CD31 staining, indicating that TRPA1 is found in ECs. (**c**, upper panel): analysis of CD31 levels by qPCR in a fraction of CD31-enriched cell population isolated from the mouse retina (CD31+) compared to the depleted cell population (CD31−); (**c**, lower panel): analysis of TRPA1 mRNA expression by qPCR in a fraction of CD3-enriched cell population isolated from the mouse retina (CD31+) compared to the depleted cell population (CD31−). Values are expressed relative to actin expression. Statistical significance *: *p*-value < 0.05, **: *p*-value < 0.005 and ***: *p*-value < 0.0005 vs. control. (**d**) Subretinal injection of AITC or HC030031 did not alter the overall development of the vascular plexus 3 days after treatment, but the front of growing vasculature became uneven after injection of the TRPA1 agonist AITC in the subretinal space. (**e**, upper panel) This was illustrated by a significant increase of 26.8% in the length of sprouts (white bars): quantification of tip cell length during mouse retinal angiogenesis shows that compared to that in controls, tip cell length increased significantly 3 days post-intravitreous injection of 200 µM AITC. (**e**, lower panel): Treatment with TRPA1 inhibitor (200 µM HC030031) induced a decrease in tip cell length. Cell lengths were either classified into three groups according to ranges (<70 µm, 70–150 µm, >150 µm) or considered altogether (total). Bars represent the mean ± SEM of each length range or total grouping. (**f**, upper panel): Percentage of tip cells with each length range shows that treatment with AITC induces a decrease in the number of short tip cells (<70 µm) and an increase in the number of medium and long tip cells (>70 µm). (**f**, lower panel): Inhibition of TRPA1 with HC030031 induces the opposite effect, increasing the number of short tip cells and decreasing the number of medium and long tip cells.

**Figure 8 cancers-11-00956-f008:**
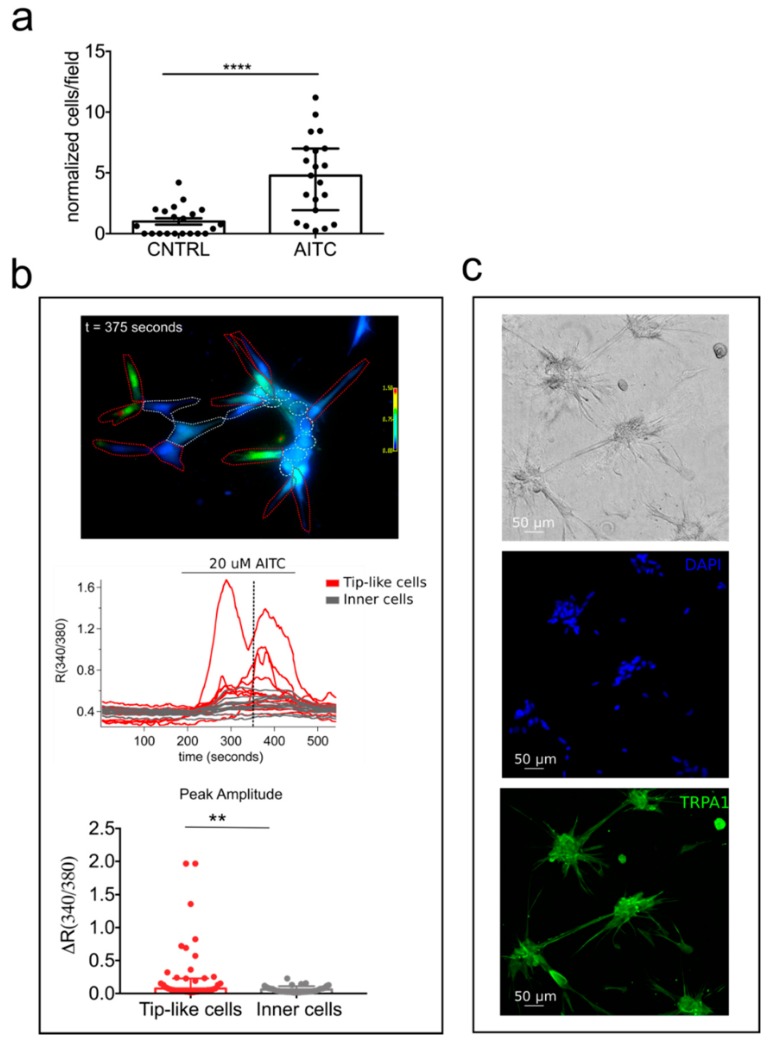
TRPA1 is implicated in PTEC chemotaxis and function in tip-like cells. (**a**) hTERT PTEC were plated in the upper chamber of 8 µm-pore culture inserts in the presence or absence of 20 μM AITC. AITC significantly increased hTERT PTEC chemoattraction through the membrane, quantified 24 h after plating. Each dot indicates cell number in a single field normalized to hTERT PTEC cell count; results are presented as normalized values against CNTRL hTERT PTEC (median ± 95% CI of normalized cells in the field cells from 4 independent experiments). Statistical significance ****: *p*-value < 0.0001 (Mann-Whitney test). (**b**) TRPA1-mediated Ca^2+^ responses in migrating (tip-like) versus non-migrating cells (inner cells). Upper panel: representative field showing 20 μM AITC-induced Ca^2+^ responses in hTERT PTEC plated on Matrigel/collagen substrate. Representative image in pseudocolor showing regions of interest (ROIs) representing “tip-like” cells (red ROIs) or “inner” cells (gray ROIs) at the peak of AITC-induced response (*t* = 375 seconds as indicated in panel c, dotted line). Middle panel: each trace represents the ratio (340/380 nm) of a single cell in the field from one representative experiment (*n* = 5 experiments). Lower panel: bar plot showing peak amplitude of AITC-mediated Ca^2+^ responses (median ± 95% CI of different cells in the field from 5 independent experiments). Statistical significance and **: *p*-value < 0.01 (Mann-Whitney test). (**c**) Representative images of hTERT PTEC plated on Matrigel/collagen substrate showing phase contrast (upper panel), DAPI (middle panel) and TRPA1 expression in green (lower panel).

**Table 1 cancers-11-00956-t001:** Primers for *trp* channels were designed on the functional region of the channels. Primer3 was used to select the best primer couple for each gene. Primer couples were chosen in order to detect all functional variants described in NCBI.

TRP Channel		PCR Product Length (nt)
Gene	NCBI Reference Sequence	
*trpv1*	NM_080704.3	101
*trpv2*	NM_016113.4	94
*trpv3*	NM_001258205.1	118
*trpv4*	NM_021625.4	99
*trpv5*	NM_019841.6	114
*trpv6*	NM_018646.5	118
*trpc1*	NM_001251845.1	118
*trpc3*	NM_001130698.1	112
*trpc4*	NM_016179.2	90
*trpc5*	NM_012471.2	111
*trpc6*	NM_004621.5	95
*trpc7*	NM_020389.2	110
*trpm1*	NM_001252020.1	91
*trpm2*	NM_003307.3	100
*trpm3*	NM_020952.4	113
*trpm4*	NM_017636.3	93
*trpm5*	NM_014555.3	117
*trpm6*	NM_017662.4	117
*trpm7*	NM_017672.5	109
*trpm8*	NM_024080.4	113
*trpa1*	NM_007332.2	95
*trpp2-pkd2*	NM_000297.3	110
*trpp3-pkd2l1*	NM_016112.2	110
*trpml1-mcoln1*	NM_020533.2	117
*trpml2-mcoln2*	NM_153259.3	91
*trpml3-mcoln3*	NM_018298.10	96

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
