# Peer review of "Transient Receptor Potential Channel Expression Signatures in Tumor-Derived Endothelial Cells: Functional Roles in Prostate Cancer Angiogenesis"

_cancers, 2019, doi:10.3390/cancers11070956_

Round 1
Reviewer 1 Report
The authors addressed adequately all my concerns
Reviewer 2 Report
Regarding to my suggestion whether TRPA1 is especially-expressed in elongating tip cells and TRPA1 activator itself act as a chemoattractant for EC tip cells, the authors show interesting results in Figure 8. AITC induces migration of hTERT PTEC cells and AITC-induced increase in [Ca2+]i is locally higher in edges of the vessel-like hTERT PTEC cells. Validation of hTERT PTEC cells as tumor-derived EC model (Figure 4) and rearrangement of Figure 6C are also convincing. Thus, this paper is appropriately revised.
This manuscript is a resubmission of an earlier submission. The following is a list of the peer review reports and author responses from that submission.
Round 1
Reviewer 1 Report
Bernadini et al., comprehensively studied the expression of all ion channels of the trp family in three endothelial primary lines derived from patients with prostate cancer and comparted their expression to that of a number of “normal” endothelial cells and endothelial cells derived from breast or renal cancers. Moreover, the authors extensively characterized the contribution of the three main trp channels that were found deregulated in their screen in different aspects of endothelial cell biology relevant to cancer angiogenesis in vitro and in vivo.
This is an interesting study that potentially provides important and novel information relevant to trp contribution in cancer angiogenesis. There are however a number of deficiencies that in my opinion, need to be addressed prior to publication. Please see details below
Major comments
Generally, the authors propose that TRPA1, TRPV2 and TRPC3 upregulation and TRPC6 downregulation is a “prostate-specific” signature in tumour-associated ECs. I find this statement rather strong. Firstly, this is based primarily on the expression profile of only three tumour-derived endothelial cells lines and immunohistochemistry from 10 patient biopsies. I believe that this is a strong statement to make based on so few samples that especially in the case of TRPC3 and TRPV2 show rather variable pattern of expression. Secondly, TRPA1, the channel that its expression is more significantly deregulated in the author’s study is also involved in developmental angiogenesis in the mouse retina (Figure 6). Consequently, I suggest that the authors tone down this statement throughout, especially in the first paragraph of the discussion.
Another important issue throughout is that although the authors in this and other studies make a convincing case that in ECs, depending on their origin there is are significant differences in trp expression they keep switching their controls. For example, in Figure 1 and Firure S2 mRNA expression or protein expression is compared to the prostate-derived HPrMEC (correctly). However, in figures 4C, Figure 5, Figure 3b and Figure 3S dermal microvascular endothelial cells are used as the control, without any explanation in the text why. This is confusing and also dilutes the author’s main message.
More specifically:
1) In figure 1 the authors profile the expression of all trp ion channels in their 3 prostate-derived primary endothelial cell lines and compare it to that of HPrMEC. This is clear and interesting. However, the western blot experiments, which in my opinion are very important to show that the message is translated into protein are relegated in the supplement (Figure 2S). Moreover, the western blots presented do not compare all tumour derived cell lines to HPrMECs and are spliced. At a minimum, the authors need to provide in the supplement a scan of the whole film in order to show that their bands are taken from the same experiment. Ideally, the authors need to provide new western blots (non-spliced) comparing TRPA1, TRPC3, TRPV2 and TRPC6 protein expression in PTEC1, PTEC2 and PTEC3 to HPrMEC.
2) The authors provide comprehensive information comparing mRNA levels of trp channels to that of HPrMEC and partial information regarding the protein expression profiles (see also Major comment 1). It would be informative if the authors also compare Ca2+ traces between PTEC1, PTEC2, PTCE3 versus the primary prostate ECs HPrMEC.
3) In Figure 3 the authors immortalize their PTCE cell line and then show that their immortalized cells retain (at passge7) some of their endothelial features when compared to the primary PTECs. The authors need to also show that the expression of TRPA1, TRPV2, TRPC3 and TRPC6 at the mRNA level is similar to that of primary PTECs.
4) In Figure 3b and Figures 3Sb-3Sf the authors compare Ca2+ traces and protein expression between HMECs and the immortalized cell line hTERT PTEC. I believe this is not correct since they compare an immortalized cell line derived from the prostate (hTERT PTEC) versus a primary cell line derived from the skin (HMEC). At a minimum this needs to be mentioned in the text and potential “artifacts” due to immortalization and the different origin of the cells highlighted. Ideally, the authors need to either compare (especially in figure 3b) PTEC1-3 with HPrMECs or if primary PTECs are no longer available to immortalize HMECs or HPrMECs with hTERT and use them as controls in these experiments.
5) Similarly in Figure 4C HMECs are compared to hTERT PTECs. This is not clear in the figure legend or the figure itself and needs to be changed. Ideally, PTEC1-3 needs to be compared to the prostate derived HPrMECs. As in comment 4 an immortalized primary EC cell line, preferably from the prostate, could also be used.
6) Again in Figure 5, at a minimum it should be made clear that hTERT PTECs are compared to primary HMECs in the figure and in the legend. Ideally, the experiment needs to be repeated either with primary PTEC1-3 versus HPrMECs or comparing PTEC2 (low TRPC3 expression ) to PTEC3 (high TRPC3 expression). If primary PTECs are no longer available an immortalized primary EC cell line could be used.
Minor comments
1) The discussion is rather long and wordy. If possible, needs to be shortened.
2) Line 565 please add the Addgene plasmid number and if appropriate the relevant reference.
3) Line 877 what is AR?
Author Response
Reviewer 1
We have answered all the points raised by the Reviewer and conducted new experiments including qPCR, immunoblot of primary cell and cell lines, immune cytochemistry, tubulogenesis assays, Ca2+ imaging on primary cells and tubules. The data were added as 3 new whole figures (Fig. 4, 8 and S2) as well as panels C, D and E in Figure 2. We thank the Reviewer for her/his critical reading of the manuscript and valuable comments, which have helped us to improve it. The Reviewers’ comments have been considered and the manuscript has been revised accordingly using word tracking system. The answers to the Reviewers’ comments can be found here below marked in green.
Major comments
Generally, the authors propose that TRPA1, TRPV2 and TRPC3 upregulation and TRPC6 downregulation is a “prostate-specific” signature in tumour-associated ECs. I find this statement rather strong. Firstly, this is based primarily on the expression profile of only three tumour-derived endothelial cells lines and immunohistochemistry from 10 patient biopsies. I believe that this is a strong statement to make based on so few samples that especially in the case of TRPC3 and TRPV2 show rather variable pattern of expression. Secondly, TRPA1, the channel that its expression is more significantly deregulated in the author’s study is also involved in developmental angiogenesis in the mouse retina (Figure 6). Consequently, I suggest that the authors tone down this statement throughout, especially in the first paragraph of the discussion.
The selected channels were specific for prostate among TEC derived from prostate, breast and kidney carcinomas. It is true that this is not representative for all TECs and screening of other carcinomas could reveal the implication of these 3 channels in other TEC signatures. We have therefore changed the term “prostate-specific” into “prostate-associated” through the whole document.
Concerning TRPA1 we fully agree that TRPA1 is not a modulator of vessel morphogenesis specifically for cancer angiogenesis. In this regard we state in the discussion “Finally, we demonstrate that TRPA1 is an important positive modulator of vessel morphogenesis, activating sprouting angiogenesis in vivo and EC migration and tubulogenesis in vitro.”
Another important issue throughout is that although the authors in this and other studies make a convincing case that in ECs, depending on their origin there is are significant differences in trp expression they keep switching their controls. For example, in Figure 1 and Firure S2 mRNA expression or protein expression is compared to the prostate-derived HPrMEC (correctly). However, in figures 4C, Figure 5, Figure 3b and Figure 3S dermal microvascular endothelial cells are used as the control, without any explanation in the text why. This is confusing and also dilutes the author’s main message.
We agree with the authors that values PTEC should refer to HprMEC since they are both primary cells and have changed the figures accordingly by adding the following panels:
- panels c, d and e in Figure 2 for Ca2+ imaging
- panel a in Figure 4 for quantitative PCR
- panels a, b, c and d in Figure 2S for immunoblotting
More specifically:
1) In figure 1 the authors profile the expression of all trp ion channels in their 3 prostate-derived primary endothelial cell lines and compare it to that of HPrMEC. This is clear and interesting. However, the western blot experiments, which in my opinion are very important to show that the message is translated into protein are relegated in the supplement (Figure 2S). Moreover, the western blots presented do not compare all tumour derived cell lines to HPrMECs and are spliced. At a minimum, the authors need to provide in the supplement a scan of the whole film in order to show that their bands are taken from the same experiment. Ideally, the authors need to provide new western blots (non-spliced) comparing TRPA1, TRPC3, TRPV2 and TRPC6 protein expression in PTEC1, PTEC2 and PTEC3 to HPrMEC.
As suggested by the Reviewer we have repeated immunoblots for TRPA1, TRPC3, TRPV2 and TRPC6 in order to have side by side HPrMEC and the PTECs. These blots are included in the revised manuscript as Figure 2S.
Please find in the attached pdf the whole films of the blots used for the figures 4 and S2.
2) The authors provide comprehensive information comparing mRNA levels of trp channels to that of HPrMEC and partial information regarding the protein expression profiles (see also Major comment 1). It would be informative if the authors also compare Ca2+ traces between PTEC1, PTEC2, PTCE3 versus the primary prostate ECs HPrMEC.
Regarding protein expression we have provided the blots asked by the Reviewer as noted in major comment 1.
We agree with the Reviewer that over-expression of a channel is not necessarily followed by an increase in function. We have therefore conducted new experiments in order to study the functional activity of the channel agonists on PTEC 1, 2 & 3 and included in the revised manuscript as panels c, d and e of Figure 2. Ca2+ imaging using the agonists of TRPA1, TRPV2 and TRPC3 shows increased Ca2+ influx for the three primary cells PTEC 1, 2 & 3, confirming thus the functional overexpression of these channels.
3) In Figure 3 the authors immortalize their PTCE cell line and then show that their immortalized cells retain (at passge7) some of their endothelial features when compared to the primary PTECs. The authors need to also show that the expression of TRPA1, TRPV2, TRPC3 and TRPC6 at the mRNA level is similar to that of primary PTECs.
As suggested by the Reviewer we have performed the qPCR of TRPA1, TRPV2, TRPC3 and TRPC6 confirming similar levels of primary PTECs with the immortalized ones (hTERT PTEC) and their difference in expression with HMEC and HPrMEC. The data were added in the revised manuscript as panel a of the Figure 4.
4) In Figure 3b and Figures 3Sb-3Sf the authors compare Ca2+ traces and protein expression between HMECs and the immortalized cell line hTERT PTEC. I believe this is not correct since they compare an immortalized cell line derived from the prostate (hTERT PTEC) versus a primary cell line derived from the skin (HMEC). At a minimum this needs to be mentioned in the text and potential “artifacts” due to immortalization and the different origin of the cells highlighted. Ideally, the authors need to either compare (especially in figure 3b) PTEC1-3 with HPrMECs or if primary PTECs are no longer available to immortalize HMECs or HPrMECs with hTERT and use them as controls in these experiments.
We thank the Reviewer for pointing out the origin of HMEC because it allows us to better clarify it. We agree with the Reviewer that comparing immortalized versus primary cells is not fair. However, HMEC used in the paper are immortalized. We are sorry we did not mentioned in the previous version, we now added the specification of the HMEC cells in the Material and Methods section:
“HMECs were obtained from the derma using an anti-CD31 antibody and MACS. HMECs from the derma were immortalized by the infection of primary cultures with a replication-defective adeno-5/SV40 virus as previously described [46,47].”
In order to have a clear vision of all cell lines used we added Ca2+ traces for HPrMEC in the revised manuscript as panels c, d and e of Figure 2 as mentioned above. We thank the Reviewer for pointing out this omission.
5) Similarly in Figure 4C HMECs are compared to hTERT PTECs. This is not clear in the figure legend or the figure itself and needs to be changed. Ideally, PTEC1-3 needs to be compared to the prostate derived HPrMECs. As in comment 4 an immortalized primary EC cell line, preferably from the prostate, could also be used.
We conserved Figure 4c (Figure 5d in the revised manuscript) as it was since TRPA1 activation was compared between PTEC with or without siRNA against TRPA1. On the other hand we have performed Ca2+ imaging experiments and added Ca2+ traces upon TRPA1 agonist stimulation in HPrMEC compared to PTEC in Figure 2c. In addition we added new Ca2+ data in Figure 4c on hTERT PTEC compared with both HprMEC and HMEC in order to validate the well-functioning of the hTERT cell line.
6) Again in Figure 5, at a minimum it should be made clear that hTERT PTECs are compared to primary HMECs in the figure and in the legend. Ideally, the experiment needs to be repeated either with primary PTEC1-3 versus HPrMECs or comparing PTEC2 (low TRPC3 expression ) to PTEC3 (high TRPC3 expression). If primary PTECs are no longer available an immortalized primary EC cell line could be used.
We thank the Reviewer for his observation and changed in the Figure 6 (Figure 5 in the old manuscript) PTEC into hTERT PTEC in order to avoid confusion. We consider that it is fait to user hTERT PTEC since we have demonstrated in Figure 4 that they show similar TRPA1 expression and Ca2+ traces upon agonist stimulation with primary PTEC.
Although PCR data show a differential TRPC3 mRNA expression between PTEC2 and PTEC3, this difference was not evident in WB (Fig2 and Fig 2S revised manuscript) and functional Ca2+ imaging experiments (Fig 2). This is why we though to perform PC3 chemotactic assays using hTERT PTEC (derived from PTEC3) in order to be able to use it instead of primary and avoid variability of PTECs.
Minor comments
1) The discussion is rather long and wordy. If possible, needs to be shortened.
We have reduced the length of discussion especially concerning the points on transcriptomic data and TRPV2 therapeutic perspectives.
2) Line 565 please add the Addgene plasmid number and if appropriate the relevant reference.
We added the reference for TRPC3 plasmid “(Addgene plasmid #25902)”
3) Line 877 what is AR?
The abbreviation is used and explained in p21 l616 “In addition, androgen receptor (AR) expression”

Reviewer 2 Report
This paper by Bernardini et al. is roughly divided into two sections. First, the authors provide the expression profile of TRP channels in endothelial cells, especially in prostate tumor-derived endothelial cells. TRPA1, TRPV2, TRPC3 and TRPC6 are identified as abnormally expressed (‘prostate-specific’) genes in endothelial cells in the prostate tumor, by comparing the mRNA and protein expression pattern between tumor-derived cells and normal endothelial counterparts (Figure 1 and 2). Secondly, the authors show the effects of these overexpressed TRPs on EC properties. Specifically, TRPV2 expression is associated with proliferation (Figure 3), TRPA1 with cell migration in wound healing assay and angiogenesis in vitro and in vivo (Figure 4 and 6), and TRPC3 with tumor cell chemoattraction (Figure 5). There findings are interesting and important, because tumor-derived endothelial cell-specific characterization of TRPs is not fully performed, while accumulating data have demonstrated the importance of TRPs in EC properties. This study may shed light on the abnormal expression of TRPs in tumor-derived endothelial cells for much better understanding of the mechanism underlying their aggressive phenotypes and tumor angiogenesis. Each experiment looks well-designed and the data is clear except for the following points.
1) While the authors use PTEC as aggressive tumor-derived endothelial cells, why do not they compare their properties with HMEC? Also, does the suppression of each ‘prostate-specific’ TRP channel suppress their aggressive phenotypes to the levels of HMEC? The connection between aggressive phenotypes of ECs in tumor and the changes of TRP expression pattern is not necessarily demonstrated directly.
2) The effect of TRPA1 agonist and antagonist on vessel sprouting is clearly shown in Figure 6d-f. Is TRPA1 expressed especially in elongating tip cells? Also, does TRPA1 activator itself act as a chemoattractant? If so, the can explain why the increase of TRPA1 expression is observed in tumor-derived endothelial cells.
3) Figure 5c is confusing because the definition of velocity (vel) in each sample is not indicated. According to Figure 5b, the vel of PC3 cells toward CONTROL looks different in each preparation, wheras the vels of CONTROL and HMEC conditioned are not so different. Also, what is the difference between the left panel and the central panel? Why are they separated?
Author Response
Reviewer 2
We have answered all the points raised by the Reviewer and conducted new experiments including qPCR, immunoblot of primary cell and cell lines, immune cytochemistry, tubulogenesis assays, Ca2+ imaging on primary cells and tubules. The data were added as 3 new whole figures (Fig. 4, 8 and S2) as well as panels C, D and E in Figure 2. We thank the Reviewer for her/his critical reading of the manuscript and valuable comments, which have helped us to improve it. The Reviewers’ comments have been considered and the manuscript has been revised accordingly using word tracking system. The answers to the Reviewers’ comments can be found here below marked in green.
1) While the authors use PTEC as aggressive tumor-derived endothelial cells, why do not they compare their properties with HMEC? Also, does the suppression of each ‘prostate-specific’ TRP channel suppress their aggressive phenotypes to the levels of HMEC? The connection between aggressive phenotypes of ECs in tumor and the changes of TRP expression pattern is not necessarily demonstrated directly.
We agree with the Reviewer that we did not develop in depth in this paper the aggressiveness of PTEC since this was one of the main messages of our previous paper Fiorio Pla et al. BMC Cancer 2014. In this paper we show that PTEC were able to migrate, organize in capillary-like structures in vitro and in vessel structures in vivo, connected with the mouse vasculature, indicating their endothelial phenotype. PTEC expressed higher AR levels than normal endothelial cells indicating the persistence of the phenotype of origin. The ability of TEC to organize into functional vessels in vivo has been previously described to be characteristic of TEC isolated from human tumors, at variance with HUVEC that undergo apoptosis.
2) The effect of TRPA1 agonist and antagonist on vessel sprouting is clearly shown in Figure 6d-f. Is TRPA1 expressed especially in elongating tip cells? Also, does TRPA1 activator itself act as a chemoattractant? If so, the can explain why the increase of TRPA1 expression is observed in tumor-derived endothelial cells.
We thank the Reviewer for raising up this really intriguing point. In order to evaluate the possible TRPA1 role in chemoattraction during sprouting angiogenesis we performed migration transwell assays stimulating hTERT PTEC cells in the presence of the channel agonist, AITC, in the lower compartment. Indeed, AITC exerted a chemoattractive effect and most interestingly when we performed Ca2+ imaging experiments on hTERT PTEC forming capillary-like structures, AITC induced a strong increase in [Ca2+]i only in cells engaged in migration (that we called here “tip-like” cells to recall “tip” cell organization in vessel morphogenesis,). These data were added in the revised manuscript as new figure 8 and are in agreement with the role of TRPA1 in vessel morphogenesis and cell migration suggesting an active role for the channel in sprouting angiogenesis and tip cells activation.
3) Figure 5c is confusing because the definition of velocity (vel) in each sample is not indicated. According to Figure 5b, the vel of PC3 cells toward CONTROL looks different in each preparation, wheras the vels of CONTROL and HMEC conditioned are not so different. Also, what is the difference between the left panel and the central panel? Why are they separated?
We agree with the Reviewer that the term CNTRL is somehow confusing and thank him for point it out. We have therefore replaced the term control with non-conditioned media in Figure 5b (Figure 6b in the revised manuscript) and changed the figure accordingly for better readability.
Reviewer 3 Report
Dear Dr. Gkika and co-authors,
I read with much interest the article entitled “Transient Receptor Potential Channel expression signature in tumor-derived endothelial cells: Functional roles in prostate cancer angiogenesis”.
This article aims at presenting a full signature of the TRPs involved in tumor angiogenesis of prostate cancer.
The paper is well written and the methodology is rather clear in most instances. Nevertheless, it raises several questions and comments that I list below:
Ø Main comments:
- The main message is blurred, biased and even undermined by detailed information about the general role of TRPs in angiogenesis:
o Are TRPA1, TRPC3, TRPC6 and TRPV2 “prostate-specific” channels? The relative mRNA levels in HMEC and GEC (Fig. 1d; Fig. 1Sc, d) seem to testify against this statement.
o The demonstration by the authors that TRPA1 is involved in vascular network formation and angiogenesis, independently from the cancer context and independently from the prostate environment, is, to my point of view totally detrimental to the strength of the paper. Even though this experiment is neatly done, it shatters the whole previous demonstration of specific signature of TRPs in prostate tumor ECs and excludes these channels from being used as therapeutic targets, as is proposed by the authors as a conclusion of the paper.
- Figure 1 presents four so-called “prostate-specific” TRP genes deregulated in TEC during PCa angiogenesis. Why does TRPC6 disappear from the following analysis? Do the authors pretend that downregulation of a gene is not important in these tumor-linked phenomena or is there a reason why this gene has been left aside?
- Figure 2S demonstrates that the 3 PTEC primary cells do not react homogeneously as for the overexpression or downregulation of the analyzed TRPs. The authors should present the immunoblots for all 3 PTECs in order to determine whether the so-called signature has a real meaning or not. What is the functional meaning if not all the cells react the same? Are there several types of PCas, with different aggressiveness depending upon the expression of one or several of these channels?
- In the same vein, the authors do not present any electrophysiological measurement showing that the channels are in the active form. Calcium measurements are only presented to testify that transfection was correct. Are these channels necessary per se, or is their action electrically mediated? What is their mechanism of action?
Ø Minor comments:
- In Figure 3a, p values are given in the figure legend, but no significance sign is present on the plot. Are the results really significant?
- The paper would gain in power of conviction if some of the representative immunoblots were neater.
As a whole I recommend refocusing the paper to only consider TRPs as important elements of the neo-angiogenesis processes in healthy organs or in such tumors such as in prostate cancer.
Author Response
Reviewer 3
We have answered all the points raised by the Reviewer and conducted new experiments including qPCR, immunoblot of primary cell and cell lines, immune cytochemistry, tubulogenesis assays, Ca2+ imaging on primary cells and tubules. The data were added as 3 new whole figures (Fig. 4, 8 and S2) as well as panels C, D and E in Figure 2. We thank the Reviewer for her/his critical reading of the manuscript and valuable comments, which have helped us to improve it. The Reviewers’ comments have been considered and the manuscript has been revised accordingly using word tracking system. The answers to the Reviewers’ comments can be found here below marked in green.
Ø Main comments:
- The main message is blurred, biased and even undermined by detailed information about the general role of TRPs in angiogenesis:
o Are TRPA1, TRPC3, TRPC6 and TRPV2 “prostate-specific” channels? The relative mRNA levels in HMEC and GEC (Fig. 1d; Fig. 1Sc, d) seem to testify against this statement.
In figure 1d, we agree with the Reviewer that there is some heterogeneity especially regarding TRPC3 expression. However, this particular situation was true only for GEC and could be explained by the peculiar nature of these endothelial cells derived from renal glomeruli. On the other hand, if we compare the PTEC versus the other tumors and all the other healthy ECs, the differences in trpc3 expression is dramatic (about 40 times). In figure 1Sc, we see that 5 genes (trpc1, trpc4, trpm1, trpm7 and trpml1) are indeed not prostate associated since they are expressed not only in HPrMEC but also in GEC (all of them) and in HMEC (trpc1 and trpm1).
The term prostate-specific refers to the deregulation in gene expression which is specific in prostate ECs and TECs, as mentioned at the end of the introduction “We identified four ‘prostate-specific’ genes whose expression is significantly and specifically deregulated in prostate TECs relative to normal ECs”. However, it should be noted that the selected channels were specific for prostate among TEC derived from prostate, breast and kidney carcinomas. It is true that this is not representative for all TECs and screening of other carcinomas could reveal the implication of these 3 channels in other TEC signatures. We have therefore changed the term “prostate-specific” into “prostate-associated” through the whole document.
o The demonstration by the authors that TRPA1 is involved in vascular network formation and angiogenesis, independently from the cancer context and independently from the prostate environment, is, to my point of view totally detrimental to the strength of the paper. Even though this experiment is neatly done, it shatters the whole previous demonstration of specific signature of TRPs in prostate tumor ECs and excludes these channels from being used as therapeutic targets, as is proposed by the authors as a conclusion of the paper.
TRPA1 shows a dramatic upregulation in TECs compared to healthy EC which also leads to a dramatic increase in TRPA1 activity compared with healthy EC as revealed by Ca2+ imaging experiments (Fig 2 and fig 4). On this basis TECS would respond strongly to an anti-cancer molecule targeting the channel. In addition we studied further the role of TRPA1 in vessel morphogenesis and cell migration in the revised manuscript. In order to evaluate the possible TRPA1 role in chemoattraction during sprouting angiogenesis we performed migration transwell assays stimulating hTERT PTEC cells in the presence of the channel agonist, AITC, in the lower compartment. Indeed, AITC exerted a chemoattractive effect and most interestingly when we performed Ca2+ imaging experiments on hTERT PTEC forming capillary-like structures. AITC induced a strong increase in [Ca2+]i only in cells engaged in migration (that we called here “tip-like” cells to recall “tip” cell organization in vessel morphogenesis,). These data were added in the revised manuscript as figure 8 and suggest an active role for the channel in sprouting angiogenesis and tip cells activation.
On the other hand, TRPA1 channel expression is not restricted to prostate TEC: it has been previously shown to be selectively expressed in cerebral arteries where it plays a role in vasodilation. Indeed, Ca2+ influx via endothelial TRPA1 channels elicits vasodilation in cerebral arteries by a mechanism involving Ca2+-activated K+ channels and inwardly rectifying K+ channels in rat myocytes. We therefore need to specify that the term “prostate-specific” (now changed in “prostate- associated) does not refer to a restricted expression of the channel by it rather reflected a specific deregulation in PTEC as compared with the healthy counterpart (HPrMEC).
- Figure 1 presents four so-called “prostate-specific” TRP genes deregulated in TEC during PCa angiogenesis. Why does TRPC6 disappear from the following analysis? Do the authors pretend that downregulation of a gene is not important in these tumor-linked phenomena or is there a reason why this gene has been left aside?
We agree with the Reviewer that gene downregulation is not less important than upregulation. We have chosen to leave out TRPC6 from further characterization since this channel showed variable expression at the protein level in PTEC, as western blotting analysis showed no change in TRPC6 expression at the protein level in PTEC relative to that in HPrMEC (Figure 2Sd). We clearly indicated this reasoning in the revised manuscript at the end of the paragraph entitled: “2.2. TRPA1, TRPV2, TRPC3 and TRPC6 expression in PCa patients”.
Moreover over-expression of a channel is not necessarily followed by an increase in function. We have therefore conducted new experiments in order to study the effect of the channel agonists on PTEC 1, 2 & 3 and included in the revised manuscript as panels c, d and e of Figure 2. Ca2+ imaging using the agonists of TRPA1, TRPV2 and TRPC3 shows increased Ca2+ influx for the three primary cells PTEC 1, 2 & 3, confirming thus the functional overexpression of these channels.
- Figure 2S demonstrates that the 3 PTEC primary cells do not react homogeneously as for the overexpression or downregulation of the analyzed TRPs. The authors should present the immunoblots for all 3 PTECs in order to determine whether the so-called signature has a real meaning or not. What is the functional meaning if not all the cells react the same? Are there several types of PCas, with different aggressiveness depending upon the expression of one or several of these channels?
As suggested by the Reviewer we have repeated immunoblots for TRPA1, TRPC3, TRPV2 and TRPC6 in the three PTEC primary cells and compared to HPrMEC and the PTECs. These blots are included in the revised manuscript as Figure 2S. Expression of TRPA1, TRPC3, TRPV2 is indeed increased in the three PTECs, however we cannot conclude on the TRPC6 expression deregulation.
PTEC cell were isolated from an homogenous PCa sample population, since the 3 patients were from 57 to 59 years old, underwent radical prostatectomy without prior treatment and samples were estimated of Gleason 9. The data generated from PTEC isolated where validated by the immunohistochemistry experiments shown in Figure 2, for which tissue samples were obtained from 10 patients treated by radical prostatectomy for prostate cancer.
- In the same vein, the authors do not present any electrophysiological measurement showing that the channels are in the active form. Calcium measurements are only presented to testify that transfection was correct. Are these channels necessary per se, or is their action electrically mediated? What is their mechanism of action?
We agree with the Reviewer that over-expression of a channel is not necessarily followed by an increase in function. We have therefore conducted new experiments in order to study the effect of the channel agonists on PTEC 1, 2 & 3 and included in the revised manuscript as panels c, d and e of Figure 2. Ca2+ imaging using the agonists of TRPA1, TRPV2 and TRPC3 shows increased Ca2+ influx for the three primary cells PTEC 1, 2 & 3, confirming thus the functional overexpression of these channels.
Ø Minor comments:
- In Figure 3a, p values are given in the figure legend, but no significance sign is present on the plot. Are the results really significant?
We thank the Reviewer for pointing out this omission, we have now included the significance in the figure.
- The paper would gain in power of conviction if some of the representative immunoblots were neater.
As suggested by the Reviewer we have run new protein gels and immunoblotted them for TRPA1, TRPC3, TRPV2 and TRPC6 in order to have side by side HPrMEC and the PTECs. These blots are included in the revised manuscript as Figure 2S as well as hTERT compared to HMECs and HPrMECS in new Figure 4b.
As a whole I recommend refocusing the paper to only consider TRPs as important elements of the neo-angiogenesis processes in healthy organs or in such tumors such as in prostate cancer.
We thank the Reviewer for her/his recommendation. The paper is indeed focusing on tumor angiogenesis since data were generated in cells coming from cancerous and healthy prostate. In this regard we have conducted 2D tubulogenesis (Figure 7) and transwell chemoattractance (Figure 8) assays. The retina experiments were solely used as a tool of in vivo angiogenesis assay rather for its physiological significance.